# TextME: Bridging Unseen Modalities Through Text Descriptions

Soyeon Hong [1]   Jinchan Kim [1]   Jaegook You [1]   Seungtaek Choi [2]   Suha Kwak [3]   Hyunsouk Cho [4]

## Abstract

Expanding multimodal representations to novel modalities is constrained by reliance on large-scale paired datasets (e.g., text–image, text–audio, text–3D, text–molecule), which are costly and often infeasible in domains requiring expert annotation such as medical imaging and molecular analysis. We introduce **TextME**, to the best of our knowledge the first modality expansion framework based on text-only training, projecting diverse modalities into LLM embedding space as a unified anchor. Our approach exploits the geometric structure of pretrained contrastive encoders to enable zero-shot cross-modal transfer using only text descriptions, without paired supervision. We empirically validate that such consistent modality gaps exist across image, video, audio, 3D, X-ray, and molecular domains, demonstrating that text-only training can preserve substantial performance of pretrained encoders. We further show that our framework enables emergent cross-modal retrieval between modality pairs not explicitly aligned during training (e.g., audio-to-image, 3D-to-image). These results establish text-only projection training as a practical alternative to paired supervision for modality expansion. The code is available at `https://soyeonhh.github.io/TextME/`.

## 1. Introduction

Modality expansion, which aligns heterogeneous data modalities into a unified embedding space, has emerged

[1]Department of Artificial Intelligence, Ajou University, Suwon, South Korea [2]Division of Language & AI, Hankuk University of Foreign Studies, Seoul, Korea [3]Graduate School of AI, POSTECH, Pohang, Korea [4]Department of Software, Ajou University, Suwon, South Korea. Correspondence to: Seungtaek Choi <seungtaek.choi@hufs.ac.kr>, Suha Kwak <suha.kwak@postech.ac.kr>, Hyunsouk Cho <hyunsouk@ajou.ac.kr>.

*Proceedings of the 43rd International Conference on Machine Learning*, Seoul, South Korea. PMLR 306, 2026. Copyright 2026 by the author(s).

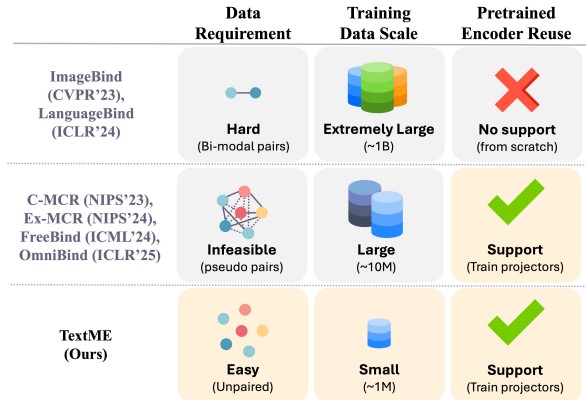

*Figure 1.* **Comparison of modality expansion approaches**. Unlike prior methods that require large-scale paired data or pseudo-pair construction through overlapping encoders, **TextME** achieves modality expansion using only unpaired text descriptions while reusing pretrained encoders.

as a core challenge in multimodal representation learning (Baltrušaitis et al., 2018; Manzoor et al., 2023; Liang et al., 2024; Yuan et al., 2025). Recent approaches leverage large-scale paired datasets to project diverse modalities—such as images, audio, and 3D point clouds—into shared semantic spaces where equivalent content maintains proximity (Zhang et al., 2023a; Han et al., 2023; Zhu et al., 2023; Lyu et al., 2024; Guo et al., 2023). While text–image and text–audio corpora have enabled remarkable progress in vision–language (Radford et al., 2021; Jia et al., 2021) and audio–language modeling (Wu et al., 2023; Manco et al., 2022), extending this paradigm to specialized domains proves prohibitively expensive or infeasible. Medical imaging requires costly expert annotations while navigating privacy constraints (Wang et al., 2025; Ziller et al., 2021), molecular analysis demands complex domain-specific representations (Xiao et al., 2024), and 3D modeling necessitates labor-intensive curation (Deitke et al., 2023). Consequently, the scalability of modality expansion remains fundamentally limited by the availability of paired supervision.

Recent methods reduce computational costs by reusing pretrained encoders through lightweight projection networks (Wang et al., 2023b; Zhang et al., 2024b; Wang et al., 2024a;b), yet they still require constructing semantically aligned pseudo pairs across all target modalities through overlapping encoders. Meanwhile, prior work has revealed

that contrastive encoders exhibit a consistent modality gap—a systematic offset between text and modality embeddings—that can enable cross-modal transfer via simple geometric operations (Liang et al., 2022; Zhang et al., 2023b; 2024a). However, these studies have primarily focused on analyzing the gap in vision-language models or mitigating the gap within paired-data settings; whether this geometric property can be exploited to eliminate the need for paired supervision altogether remains unexplored.

In this work, we demonstrate that the modality gap can enable modality expansion without paired supervision. We propose **TextME**, a framework that projects modality-specific embeddings into LLM embedding space as a unified semantic anchor by applying precomputed offset corrections derived from the gap structure. As illustrated in Figure 1, unlike prior methods that require large-scale bi-modal pairs or pseudo-pair construction through overlapping encoders, our proposed framework achieves modality expansion using only unpaired text descriptions—with substantially reduced data requirements while fully leveraging pretrained encoders through lightweight projectors.

We evaluate the framework across six diverse modalities—image, video, audio, 3D point clouds, X-ray, and molecules—on both cross-modal retrieval and zero-shot classification tasks. Our experiments demonstrate that **TextME** achieves competitive performance relative to paired-data methods and, notably, enables emergent cross-modal capabilities between modality pairs never observed during training, such as audio-to-3D and molecule-to-image retrieval. These results suggest that text modality can create meaningful semantic bridges across arbitrary modalities without explicit cross-modal supervision. To better understand the variation in performance across modalities, we further analyze the geometric properties of each encoder and find that the consistency of gap-content orthogonality correlates with downstream performance, providing insight into when text-only expansion is most effective. Our project page and code are publicly available at https://soyeonhh.github.io/TextME/.

Our contribution is three-fold:

- We propose **TextME**, a modality expansion framework based on text-only training that exploits modality gap geometry to learn cross-modal projections using only text descriptions, eliminating the need for paired multimodal supervision during training.
- We investigate LLM embedding space as a unified anchor for modality expansion and compare it against multimodal encoder representations, analyzing their varying effectiveness across tasks and modalities.
- We empirically validate the framework across six diverse modalities, demonstrating competitive performance on retrieval and classification tasks and identify-

ing encoder characteristics that predict when text-only expansion is most effective.

## 2. Preliminaries

### 2.1. Problem Formulation

Modality expansion aims to integrate pretrained modality-specific encoders into a unified semantic space where similar concepts maintain proximity regardless of their source modality. Let $\mathcal{M} = \{m_1, \ldots, m_k\}$ denote a set of target modalities to be aligned. For each modality $m \in \mathcal{M}$, a pretrained contrastive encoder consists of a text branch $E_m^{\text{text}} : \mathcal{T} \to \mathbb{R}^{d_m}$ and a modal branch $E_m^{\text{modal}} : \mathcal{X}_m \to \mathbb{R}^{d_m}$, where $\mathcal{T}$ is the space of text descriptions and $\mathcal{X}_m$ is the input space for modality $m$. Our objective is to learn projection networks $P_m : \mathbb{R}^{d_m} \to \mathbb{R}^{d_h}$ that map modal embeddings into a shared $d_h$-dimensional anchor space.

Existing methods require instance-level paired data $\{(x_i, t_i)\}_{i=1}^N$ of modal inputs $x_i \in \mathcal{X}_m$ and text descriptions $t_i \in \mathcal{T}$ to train cross-modal projections (Han et al., 2023; Zhu et al., 2023; Lyu et al., 2024). Recent approaches connect multiple pretrained encoders via overlapping modalities: given encoders pretrained on modality pairs $(\mathcal{A}, \mathcal{B})$ and $(\mathcal{B}, \mathcal{C})$, they leverage data from the shared modality $\mathcal{B}$ to align encoder spaces, enabling transfer to non-overlapping pairs $(\mathcal{A}, \mathcal{C})$ (Wang et al., 2023b; Zhang et al., 2024b; Wang et al., 2024a;b). This requires modality overlap across all target encoders. In this work, we consider a more practical scenario: learning projection networks independently for each modality using only unpaired text descriptions $\{t_i\}_{i=1}^N$, without requiring cross-encoder alignment or access to target modality samples.

### 2.2. Modality Gap and Interchangeable Space

Prior work has shown that contrastive encoders trained with objectives such as InfoNCE exhibit a systematic offset between text and modal embedding spaces (Liang et al., 2022; Zhang et al., 2023b; 2024a). For each encoder $E_m$, the modality gap is characterized by the difference between the centroids of modal and text embeddings:

$$\Delta_m = \mu_m^{\text{modal}} - \mu_m^{\text{text}}, \tag{1}$$

where $\mu_m^{\text{modal}} = \mathbb{E}[E_m^{\text{modal}}(x)]$ and $\mu_m^{\text{text}} = \mathbb{E}[E_m^{\text{text}}(t)]$ denote the expected embeddings over their respective distributions. This gap presents a fundamental challenge for text-only training, as projection networks learned from text embeddings cannot directly transfer to modal embeddings that occupy a different region of the space.

**Interchangeable Space via Centering.** A key observation from Zhang et al. (2024a) is that this challenge can be addressed through independent centering operations. Con-

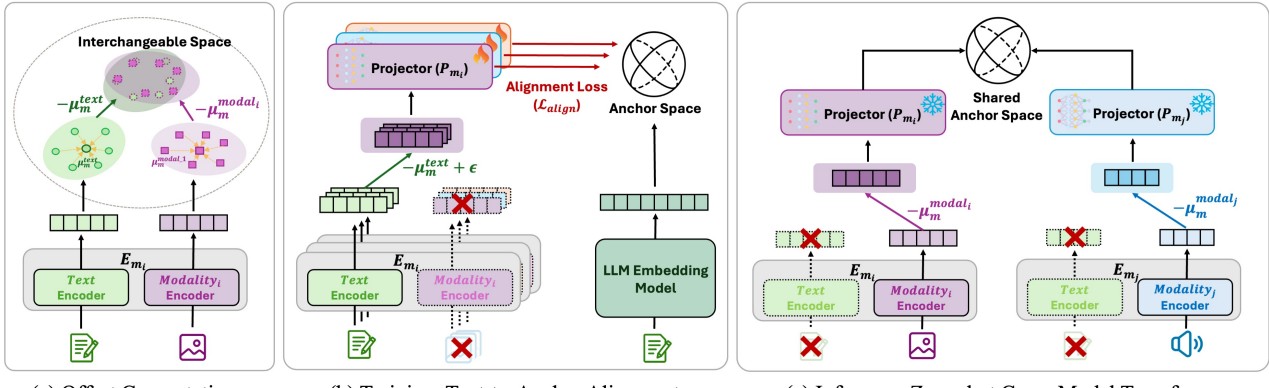

|     |     |     |
| --- | --- | --- |
| (a) Offset Computation | (b) Training: Text-to-Anchor Alignment | (c) Inference: Zero-shot Cross-Modal Transfer |

*Figure 2.* **Overview of the TextME pipeline**. (a) Offset computation estimates modality-specific centroids from unpaired samples, creating an interchangeable space where centered text and modal embeddings become functionally equivalent. (b) During training, projection networks are learned by aligning centered text embeddings with a unified LLM anchor space, requiring only text descriptions. (c) At inference, centering modal embeddings with the precomputed offset enables zero-shot cross-modal transfer without paired supervision.

sider a semantically matched pair $(t, x)$ with embeddings $e_t = E_m^{\text{text}}(t)$ and $e_x = E_m^{\text{modal}}(x)$. Although these embeddings differ due to the modality gap, subtracting their respective centroids yields centered embeddings

$$\hat{e}_t = e_t - \mu_m^{\text{text}}, \quad \hat{e}_x = e_x - \mu_m^{\text{modal}}, \tag{2}$$

that satisfy $\hat{e}_t \approx \hat{e}_x$ for semantically corresponding pairs. That is, centering removes the modality-specific bias while preserving the shared semantic content, creating an *interchangeable space* where text and modal embeddings become functionally equivalent (An et al., 2025). This property enables projection networks trained on centered text embeddings to generalize to centered modal embeddings at inference time, forming the basis of our text-only training approach. While this property has been analyzed within single bimodal encoders under paired-data settings (Zhang et al., 2024a), whether it can be exploited to bridge independently pretrained encoders without paired supervision remains unexplored.

## 3. TextME: Text-only Modality Expansion

We present **TextME**, a framework that enables modality expansion using only text descriptions by exploiting the geometric properties of pretrained contrastive encoders. Figure 2 illustrates the overall pipeline.

### 3.1. Overview

The key insight of **TextME** is that the interchangeable space described in Section 2 allows projection networks trained on centered text embeddings to generalize to centered modal embeddings at inference time. Our framework operates in two phases. During training, we precompute modality-specific centroids and train lightweight projection networks to map centered text embeddings into a shared anchor space.

At inference, we apply the same centering operation to modal embeddings before projection, enabling zero-shot cross-modal transfer without having observed any modal samples during training.

### 3.2. Offset Computation

As established in Section 2, creating an interchangeable space requires estimating the centroids $\mu_m^{\text{text}}$ and $\mu_m^{\text{modal}}$ for each modality. We compute these centroids from representative samples:

$$\mu_m^{\text{text}} = \frac{1}{N} \sum_{i=1}^{N} E_m^{\text{text}}(t_i), \quad \mu_m^{\text{modal}} = \frac{1}{M} \sum_{j=1}^{M} E_m^{\text{modal}}(x_j), \tag{3}$$

where $\{t_i\}_{i=1}^{N} \subset \mathcal{T}$ and $\{x_j\}_{j=1}^{M} \subset \mathcal{X}_m$ are sampled independently from text and modal distributions. Unlike projection training, these samples need not be instance-level paired—only representative coverage of each distribution is required for accurate centroid estimation.

Importantly, accurate centroid estimation requires only a small number of samples. In our experiments, we find that 5K samples suffice for stable estimation across all evaluated modalities, representing less than 5% of typical paired training requirements (Zhu et al., 2023; Zhang et al., 2024b). The centroids are precomputed once and remain fixed throughout training.

### 3.3. Text-to-Anchor Alignment

Given the precomputed offsets, we train projection networks using only text descriptions from the target domain. For each modality $m$, a projection network $P_m : \mathbb{R}^{d_m} \to \mathbb{R}^{d_h}$ maps centered text embeddings into a shared anchor space.

**Anchor Space Selection.** We adopt LLM embedding space as our unified anchor rather than multimodal text encoders. While multimodal encoders such as CLIP are optimized for cross-modal matching, LLMs trained on large-scale text corpora capture richer semantic relationships that generalize across diverse domains. To assess cross-domain alignment capabilities, we analyze 3K audio-image caption pairs from FlickrNet (Senocak et al., 2018), where we generated linguistically distinct but semantically equivalent descriptions using the Gemini API (Google, 2024)—for instance, an image caption "a red sports car speeding on highway" is paired with its audio equivalent "loud engine roar with wind rushing past." As shown in Figure 3, LLM embeddings (i.e., Qwen) exhibit clearer separation between semantically equivalent and unrelated pairs (0.56 vs. 0.23–0.26 mean cosine similarity) compared to multimodal encoders, suggesting better suitability for bridging heterogeneous descriptions. This advantage is further corroborated by semantic textual similarity benchmarks, where LLM embeddings achieve Spearman correlations of 85–90 compared to 67–68 for multimodal encoders (see Appendix A for details). Based on these findings, we adopt Qwen3-Embedding (Zhang et al., 2025) as our default anchor space.

**Training Objective.** Given text descriptions $\mathcal{D}_{\text{text}} = \{t_i\}_{i=1}^N$ from the target modality domain, we train the projection network by aligning centered text embeddings with their corresponding LLM embeddings:

$$\mathcal{L}_{\text{align}} = -\frac{1}{B} \sum_{i=1}^{B} \log \frac{\exp(\text{sim}(z_i, z_i')/\tau)}{\sum_{j \in \mathcal{N}_i \cup \{i\}} \exp(\text{sim}(z_i, z_j')/\tau)} \quad (4)$$

where $z_i = P_m(\hat{e}_{t_i})$ is the projected centered text embedding with $\hat{e}_{t_i} = E_m^{\text{text}}(t_i) - \mu_m^{\text{text}}$, $z_i' = E_{\text{LLM}}(t_i)$ is the corresponding LLM embedding, and $\mathcal{N}_i$ contains hard negatives. Following recent language embedding models (Lee et al., 2024; Moreira et al., 2024; Rösch et al., 2024), we employ hard negative mining to focus training on challenging examples near the decision boundary, improving the discriminative quality of learned projections.

### 3.4. Inference

At inference time, **TextME** enables zero-shot cross-modal transfer by mapping modal embeddings into the interchangeable space. For a non-text input $x$ from modality $m$, the final embedding is computed as:

$$e_{\text{final}} = P_m(\hat{e}_x) = P_m(E_m^{\text{modal}}(x) - \mu_m^{\text{modal}}). \quad (5)$$

The centering operation transforms the modal embedding into the interchangeable space where text embeddings reside during training. As established in Section 2, centering preserves semantic relationships while removing modality-specific bias, allowing the text-trained projection network

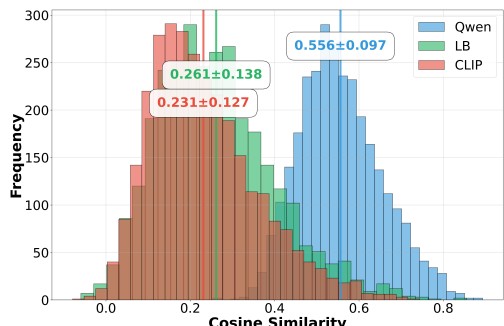

*Figure 3.* **Semantic anchoring comparison.** LLM embeddings and multimodal encoders are compared on 3K semantically equivalent cross-modal description pairs. LLM embeddings exhibit clearer separation between matched and unmatched pairs, demonstrating superior cross-domain alignment capability.

$P_m$ to process modal inputs without modification. The final embeddings are thus aligned within the unified LLM anchor space alongside text representations, enabling direct cross-modal retrieval and zero-shot classification.

## 4. Experiments

We evaluate **TextME** on cross-modal retrieval and zero-shot classification across six modalities: image, video, audio, 3D, X-ray, and molecules. Our experiments address three key questions: (1) whether text-only training can achieve competitive performance relative to paired-data methods and pretrained encoders (Section 4.2); (2) what geometric properties predict success or failure across different modalities (Section 4.3); and (3) how different design choices including anchor space selection and offset correction influence overall performance (Section 4.4).

### 4.1. Experimental Setup

**Modalities and Encoders.** We adopt LanguageBind (Zhu et al., 2023) as the text encoder for all text-to-modal retrieval and zero-shot classification tasks. For modal encoders, we adopt CLIP (Radford et al., 2021) for image, ViCLIP (Wang et al., 2023a) for video, CLAP (Elizalde et al., 2023) for audio, Uni3D (Zhou et al., 2023) for 3D, CXR-CLIP (You et al., 2023) for X-ray, and MoleculeSTM (Liu et al., 2023) for molecule. Each encoder pair is independently projected into the shared LLM anchor space for cross-modal matching. We sample 100K text descriptions per modality for projection training, with offset computation on 5K samples. Training text sources include AudioCaps (Kim et al., 2019) captions for audio, Objaverse (Deitke et al., 2023) annotations for 3D, PubChem (Kim et al., 2025) compound descriptions for molecule, CheXpert (Irvin et al., 2019) radiology reports for X-ray, and COCO (Lin et al., 2014) captions for image and video. Details of offset computation datasets are provided in Appendix C.3 (Table 15).

*Table 1.* **Zero-shot performance across all evaluation benchmarks. PPR**: Performance Preservation Ratio (%) relative to pretrained encoders. × indicates unavailable results due to missing official implementations or incompatible evaluation protocols. Bold indicates best among unpaired methods. †Our reproduction.

| | Text→X Retrieval | | | | | | | | Classification | | | | | Emergent X→X | | |
| | Image | | MSR. | Video | | Audio | | Mol. | Audio | | 3D | | X-ray | A→I | 3D→I | Data |
| | COCO | Flkr. | | MSVD | DiDe. | ACaps. | Clo. | Drug. | ASet. | ESC | MN40. | Scan. | RSNA | Flkr. | Obja. | Requirements |
|---|---|---|---|---|---|---|---|---|---|---|---|---|---|---|---|---|
| Pretrained | 48.29 | 77.70 | 37.00 | 51.06 | 31.27 | 22.47 | 16.90 | 79.19 | 9.32 | 85.20 | 67.75 | 42.21 | 52.64 | × | × | |
| *Paired-data methods* | | | | | | | | | | | | | | | | |
| LanguageBind | 44.53 | 73.42 | 45.30 | 65.22 | 36.85 | 12.42 | 11.32 | × | 18.33 | 94.00 | × | × | × | 1.52 | × | 10M pairs |
| Ex-MCR | 40.24 | 71.89 | × | × | × | 19.07 | 7.01 | × | 6.67 | 71.20 | 66.53 | 40.31 | × | 1.57 | 5.67 | 1M pairs* |
| *Unpaired-data methods* | | | | | | | | | | | | | | | | |
| Naïve | 0.01 | 0.04 | 0.00 | 0.00 | 0.00 | 0.02 | 0.04 | 10.17 | 1.14 | 2.90 | 0.81 | 3.32 | 26.36 | 0.02 | 0.00 | 0 |
| COX† | 0.02 | 0.20 | 5.10 | 0.00 | 0.10 | 0.08 | 0.11 | 7.63 | 1.26 | 2.00 | 4.05 | 2.84 | 22.53 | 0.02 | 0.00 | 10K labels |
| **TextME** | **28.63** | **51.66** | **26.40** | **45.82** | **24.10** | **15.35** | **7.81** | **34.75** | **5.80** | **77.25** | **70.86** | **42.15** | **46.59** | **1.06** | **10.27** | **100K text** |
| **PPR(%)** | 59.3 | 66.5 | 71.4 | 89.7 | 77.1 | 68.3 | 46.2 | 43.9 | 62.2 | 90.7 | 104.6 | 99.9 | 88.5 | × | × | |

*Indirect: uses overlapping modality from existing MCR spaces. TextME requires **zero paired data** and **zero labeled target data**.

**Baselines.** We compare against three categories of methods. First, we report the performance of the original *pretrained encoders* as reference points. Second, for *paired-data approaches*, we include LanguageBind (Zhu et al., 2023) and Ex-MCR (Zhang et al., 2024b), both of which perform modality expansion using fully-paired multimodal data. Third, for *unpaired-data methods*, we compare with COX† (Huang et al., 2025), which learns target modality representations from scratch without instance-level pairing but requires substantial target modality data and classification labels. We also include a Naïve baseline that simply aligns embedding dimensions via PCA without any learned projection. Unlike COX, **TextME** requires no target modality data during training.

**Evaluation.** We evaluate on three task categories: Text→X retrieval, emergent cross-modal retrieval between unseen modality pairs, and zero-shot classification. Table 1 reports results on representative benchmarks per modality, selected based on prevalence in prior work (Zhu et al., 2023; Zhang et al., 2024b). We report Recall@$k$ (R@$k$) for retrieval, MRR@$k$ for molecule retrieval following Liu et al. (2023), and Top-$k$ accuracy for classification. We define *Performance Preservation Ratio* (PPR) as the percentage of pretrained encoder performance retained by our method: PPR = (**TextME** score/Pretrained score) × 100%. Complete results across all benchmarks appear in Appendix B.

### 4.2. Does Text-Only Training Preserve Pretrained Performance?

Table 1 reports performance across all evaluation tasks. **TextME** achieves an average of 74.5% PPR across all tasks, with classification at 89.2% consistently outperforming retrieval at 65.3%, suggesting that offset-based alignment preserves categorical boundaries more effectively than fine-grained similarity structure. Among unpaired baselines, COX (Huang et al., 2025) yields substantially lower perfor-

mance, as it requires a pretrained classifier on labeled target data. Since official implementations are not publicly available, we trained this classifier from scratch on evaluation data. In contrast, our framework eliminates the need for labeled target data and paired supervision, thereby enabling direct generalization to novel modalities.

**Zero-Shot Retrieval and Classification.** Across both task categories, **TextME** substantially outperforms unpaired baselines and achieves comparable results to paired-data methods. As reported in the *Data Requirements* column of Table 1, this performance is achieved using only 100K text descriptions, compared to 1–10M paired samples required by methods such as LanguageBind and Ex-MCR. This reduction of over two orders of magnitude in supervision requirements makes modality expansion practical for specialized domains where paired annotation is prohibitively expensive. In terms of task-specific patterns, classification consistently demonstrates higher preservation than retrieval, as categorical discrimination requires only well-separated decision boundaries whereas retrieval demands fine-grained instance-level similarity that is more sensitive to distortions introduced by offset correction. Notably, 3D zero-shot classification surpasses pretrained Uni3D with 104.6% PPR on ModelNet40, while retrieval preservation varies substantially across modalities. We examine the factors underlying these variations in Section 4.3.

**Emergent Cross-Modal Capabilities.** The unified anchor space additionally enables retrieval between modality pairs not explicitly aligned during training. As reported in the Emergent X→X columns of Table 1, TextME outperforms Ex-MCR on 3D→Image despite the latter requiring paired supervision, and achieves comparable performance to paired-data methods on Audio→Image. The disparity between these two tasks is examined in Section 4.3, where we show that encoder geometric properties govern the quality of offset correction and thus cross-modal transfer.

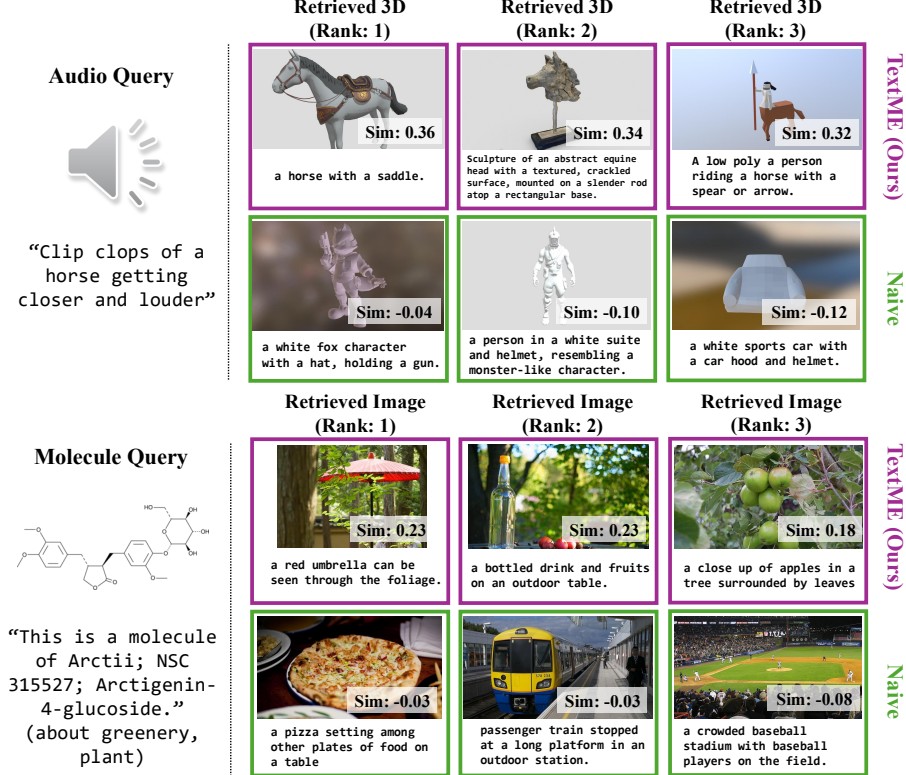

Figure 4. **Emergent cross-modal retrieval without paired supervision.** Audio queries retrieve semantically related 3D objects (top) and molecular structures retrieve contextually appropriate images (bottom). For each query, TextME (green) is compared against the Naïve baseline (gray). TextME retrieves semantically coherent results with positive cosine similarities, while Naïve yields unrelated objects with near-zero or negative similarities. These modality pairs were never seen during training, demonstrating that text-anchored alignment creates meaningful semantic bridges across arbitrary modalities.

To qualitatively examine whether the learned representations enable retrieval between modality pairs without any paired annotations, we conduct cross-modal retrieval experiments using independently collected datasets. Specifically, we sample instances from each modality and perform retrieval across disjoint modality pairs such as Audio→3D and Molecule→Image. Figure 4 presents representative results, comparing TextME against the Naïve baseline. TextME retrieves semantically coherent results with positive cosine similarities, while the Naïve baseline yields unrelated objects, confirming that the learned projections establish meaningful cross-modal correspondences.

**Representation Analysis.** We further examine how the framework preserves semantic structure by measuring Centered Kernel Alignment (CKA) (Kornblith et al., 2019) between original and projected modal embeddings. As reported in Table 2, modalities with higher downstream PPR tend to retain greater structural similarity after projection (3D at 0.904, Video at 0.866, Image at 0.832), while Molecule exhibits lower preservation at 0.779. X-ray is a no-

table exception, achieving high classification PPR (88.5%) despite the lowest CKA (0.472). We attribute this to the nature of the downstream task, as classification requires only well-separated decision boundaries rather than preservation of the full embedding geometry. Figure 5 provides complementary t-SNE visualizations for Video and Molecule, selected as the modalities with the highest and lowest retrieval PPR, respectively. Video embeddings transition from clearly separated modality clusters to overlapping distributions across the three stages, consistent with its high PPR. In contrast, Molecule embeddings lack clear modality separation at any stage, suggesting that the centering operation does not induce a meaningful structural change for this encoder.

### 4.3. When Does Text-Only Expansion Succeed?

We hypothesize that the effectiveness of text-only expansion depends on how well pretrained encoders satisfy the geometric properties underlying modality gap alignment. The re-

*Table 2.* **Structural preservation across modalities.** CKA between original and projected modal embeddings measures how much semantic structure is retained through our framework.

| Encoder | Modality | CKA (↑) |
|---|---|---|
| Uni3D | 3D | 0.904 |
| ViCLIP | Video | 0.866 |
| CLIP | Image | 0.832 |
| CLAP | Audio | 0.807 |
| MoleculeSTM | Molecule | 0.779 |
| CXR-CLIP | X-ray | 0.472 |

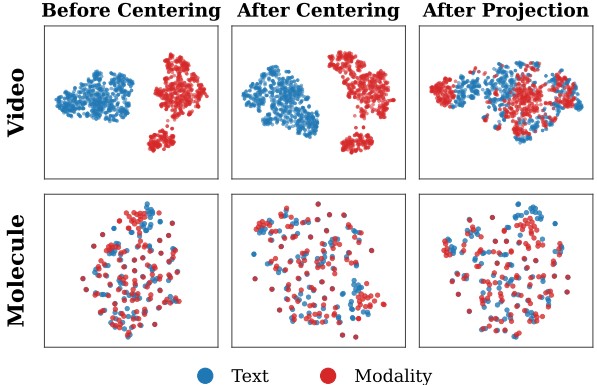

**Before Centering** **After Centering** **After Projection**

Video

Molecule

● Text ● Modality

*Figure 5.* **t-SNE visualization of embedding transformations.** We select Video and Molecule as they exhibit the highest (MSVD, 89.7%) and lowest (DrugBank, 43.9%) retrieval PPR, respectively. Video transitions from separated clusters to overlapping distributions, while Molecule shows minimal change across stages. See Appendix F for remaining modalities.

sults above support this view, revealing substantial variation in performance preservation across modalities—ranging from over 100% for 3D classification to approximately 42% for Molecule retrieval. To investigate this relationship, we measure the geometric characteristics of each encoder and examine their correlation with downstream performance.

**Geometric Properties.** Following prior work (Zhang et al., 2024a), we measure four geometric properties for each encoder using 5K paired samples: (i) **Intra-modal independence**: $\mathbb{E}[\cos(\hat{e}, \hat{\mu}_m)]$, measuring whether embeddings are statistically independent from the modality centroid; (ii) **Gap consistency**: $\cos(\Delta_m^{(k)}, \Delta_m)$, measuring whether instance-level offsets $\Delta_m^{(k)} = e_{\text{modal}}^{(k)} - e_{\text{text}}^{(k)}$ align directionally with the group-level offset $\Delta_m$; (iii) **Bounded deviation**: $\text{std}(\epsilon_k)$ where $\epsilon_k = \Delta_m^{(k)} - \Delta_m$, measuring the variance of instance-level offsets around the mean; (iv) **Gap-content orthogonality**: $|\cos(\Delta_m, r^{(p,q)})|$ where $r^{(p,q)} = e_p - e_q$, measuring whether the modality gap is independent of intra-modal semantic variations. Properties (i) and (ii) ensure that a single offset vector can characterize the modality gap, while (iii) and (iv) ensure that offset correction preserves semantic relationships.

*Table 3.* **Geometric properties of contrastive encoders.** (i) intra-modal independence, (ii) gap consistency, (iii) bounded deviation, (iv) gap-content orthogonality. †: weaker satisfaction (>0.1 for (i), <0.96 for (ii)).

| Encoder | Mod. | (i)↓ | (ii)↑ | (iii)↓ | (iv)↓ |
|---|---|---|---|---|---|
| CLIP | Image | $.28^{\dagger}_{\pm.11}$ | $.97_{\pm.00}$ | $.00_{\pm.00}$ | $.00_{\pm.11}$ |
| ViCLIP | Video | $.18^{\dagger}_{\pm.11}$ | $.98_{\pm.00}$ | $.00_{\pm.00}$ | $.00_{\pm.06}$ |
| CLAP | Audio | $.12^{\dagger}_{\pm.18}$ | $.97_{\pm.01}$ | $.00_{\pm.00}$ | $.00_{\pm.15}$ |
| Uni3D | 3D | $.07_{\pm.06}$ | $.96_{\pm.00}$ | $.00_{\pm.00}$ | $.00_{\pm.04}$ |
| CXR-CLIP | X-ray | $.37^{\dagger}_{\pm.13}$ | $.99_{\pm.00}$ | $.00_{\pm.00}$ | $.00_{\pm.06}$ |
| MoleculeSTM | Molecule | $.01_{\pm.19}$ | $.78^{\dagger}_{\pm.05}$ | $.00_{\pm.00}$ | $.00_{\pm.18}$ |

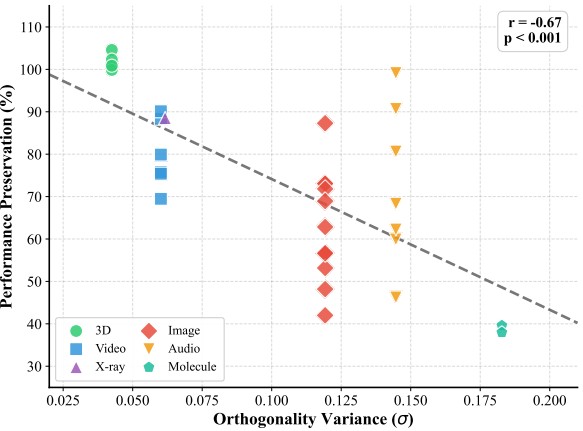

*Figure 6.* **Orthogonality variance vs. performance preservation.** Each point represents a single evaluation metric from six modalities, with tasks sharing the same encoder aligned vertically at identical variance values. Lower variance in gap-content orthogonality corresponds to higher downstream performance.

**Observations.** Table 3 demonstrates that all encoders satisfy the requirements for offset-based alignment in expectation. Gap consistency exceeds 0.96 for five of six modalities, and mean orthogonality remains near zero across all encoders, indicating that properties (ii) and (iv) hold on average. However, the degree to which these properties hold at the instance level varies substantially. We find that the variance of property (iv), gap-content orthogonality, serves as a particularly informative predictor of downstream performance. To quantify this relationship, we analyze performance preservation at the individual task level, treating each evaluation metric as a separate observation. This yields 33 data points spanning retrieval benchmarks such as Audio-Caps R@1, COCO R@5, and DrugBank MRR, as well as classification benchmarks such as ModelNet40 Top-1 and ESC-50 accuracy. Since tasks evaluated on the same encoder share identical orthogonality variance, they appear vertically aligned in Figure 6. The analysis reveals a moderate negative correlation between orthogonality variance and performance preservation, with Pearson $r = -0.67$ and $p < 0.001$. Encoders exhibiting lower variance in property (iv), notably Uni3D at $\pm0.04$ and ViCLIP at $\pm0.06$,

*Table 4.* **Effect of offset correction.** Modalities with strong gap consistency benefit substantially, while Molecule with weak consistency shows degradation.

| Modality | Benchmark | w/o offset | w/ offset | Δ |
|----------|-----------|------------|-----------|---|
| 3D | ModelNet40 | 4.05 | 70.86 | +94.30% |
| 3D | ScanObjectNN | 5.40 | 42.15 | +87.20% |
| Audio | AudioCaps | 8.68 | 15.35 | +43.50% |
| Audio | Clotho | 4.77 | 7.81 | +38.90% |
| X-ray | RSNA | 31.35 | 46.59 | +32.70% |
| Molecule | DrugBank | 36.44 | 34.75 | −4.60% |

*Table 5.* **Anchor space analysis and multi-anchor ensemble.** Individual anchors exhibit task-dependent strengths; score-level late fusion yields consistently strong performance across both task types. **Bold**: best; underline: second best.

| Anchor | Retrieval | | Classification | | |
|--------|-----------|------|---------------|-----|------|
| | ACaps R@1 | Drug. Acc@10 | MN40 Top-1 | ESC Top-1 | RSNA Top-1 |
| *Multimodal encoders* | | | | | |
| CLIP | 15.9 | 36.4 | 78.0 | **86.7** | 48.3 |
| LanguageBind | 14.5 | 29.7 | **81.1** | 74.7 | 45.0 |
| *LLM embedding models* | | | | | |
| NV-Embed-v2 | 16.2 | 26.3 | 78.3 | 79.4 | **48.6** |
| Qwen3-Embed | 15.4 | 34.8 | 79.0 | 77.2 | 46.6 |
| Ensemble | **19.2** | **41.5** | 79.7 | **86.7** | 46.9 |

achieve preservation rates consistently above 80%, whereas those with higher variance such as CLAP at $\pm 0.15$ and MoleculeSTM at $\pm 0.18$ show greater performance degradation. This pattern suggests that inconsistent orthogonality introduces variable distortions during offset correction, with fine-grained retrieval tasks affected more severely than categorical classification. We additionally note that MoleculeSTM exhibits a distinct failure mode in property (ii), as its gap consistency of only 0.78 indicates that a single offset vector inadequately characterizes the modality gap for molecular embeddings. We provide a theoretical interpretation connecting these geometric properties to the formal framework of Zhang et al. (2024a) in Appendix G.

### 4.4. How Do Design Choices Affect Performance?

**Effect of Offset Correction.** Table 4 examines the contribution of offset correction across modalities with varying gap consistency. For modalities satisfying strong gap consistency above 0.95, offset correction yields substantial improvements, with 3D classification increasing from 4.05% to 70.86% on ModelNet40 and Audio retrieval improving by 43.5% on AudioCaps. In contrast, Molecule with gap consistency of only 0.78 exhibits a slight performance degradation of 4.6%, indicating that unreliable offset estimation can introduce harmful distortions. These results suggest that practitioners should verify gap consistency before applying geometric alignment.

*Table 6.* **Effect of training data source.** Domain-specific captions substantially outperform general-purpose text across all modalities.

| Training Data | Audio R@1 | 3D Top-1 | X-ray Top-1 | Mol. MRR |
|---------------|-----------|----------|-------------|----------|
| all-NLI | 6.36 | 12.10 | 22.48 | 16.10 |
| Domain captions | 15.35 | 70.86 | 46.59 | 34.75 |
| Improvement | +141% | +485% | +107% | +116% |

**Anchor Space Selection.** Table 5 compares two categories of anchor spaces and their combination via score-level late fusion. Individual anchors exhibit a task-dependent pattern. For retrieval, LLM-based anchors such as NV-Embed-v2 outperform multimodal encoders (AudioCaps R@1: 16.2 vs. 14.5–15.9), consistent with the superior semantic textual similarity of LLM embeddings (Spearman $\rho = 85$–90) compared to multimodal encoders ($\rho = 67$–68; see Appendix A). For classification, multimodal encoders achieve the strongest results, with LanguageBind leading on ModelNet40 (81.1) and CLIP on ESC-50 (86.7), benefiting from discriminative decision boundaries acquired through vision-language contrastive pretraining. No single anchor dominates across all tasks, which motivates a multi-anchor ensemble. Score-level late fusion yields the best retrieval performance (AudioCaps 19.2, DrugBank 41.5) while matching or tying the best individual classification results (ESC-50 86.7), effectively mitigating the retrieval-classification gap identified in Section 4.2. Based on these findings, we adopt Qwen3-Embedding as the default single-anchor space for its balanced performance, while the ensemble strategy presents a promising direction for achieving task-agnostic performance without relying on a single anchor space.

**Training Data Source.** To validate whether general-purpose text corpora that have never been associated with any target modality can enable cross-modal transfer, we train projection networks using all-NLI, a corpus combining MNLI (Williams et al., 2018) and SNLI (Bowman et al., 2015) with 100K sentence pairs, following the previous work of text-only training (Xiao et al., 2025). Table 6 presents the results. Training on all-NLI yields substantially lower performance compared to domain-specific captions across all modalities, with the degradation being most pronounced for 3D, which drops from 70.86% to 12.10%. This performance gap reflects the distributional mismatch between general linguistic expressions and the specialized vocabularies characteristic of each modality domain. Nevertheless, the non-trivial transfer achieved with general-purpose text validates the paradigm, while suggesting that distributional alignment between training text and the target domain is an important factor for practitioners to consider.

## 5. Related Work

**Modality Expansion.** Contrastive learning has enabled effective multimodal alignment by projecting different modalities into shared semantic spaces (Radford et al., 2021; Jia et al., 2021). Subsequent work extends this paradigm to multiple modalities through central hubs: ImageBind (Girdhar et al., 2023) uses images as the anchor modality, while LanguageBind (Zhu et al., 2023) leverages text for broader semantic coverage. To reduce computational costs, recent methods connect frozen pretrained encoders through lightweight projectors. C-MCR (Wang et al., 2023b) and Ex-MCR (Zhang et al., 2024b) learn adapters between encoder pairs, while FreeBind (Wang et al., 2024a) and OmniBind (Wang et al., 2024b) ensemble multiple encoders per modality. However, all these approaches require instance-level paired supervision during training, which becomes prohibitive in specialized domains where paired data is scarce. **TextME** eliminates this requirement through text-only training of projection networks.

**Modality Gap Analysis.** The modality gap—a systematic offset between text and non-text embeddings in contrastive models—was first identified by Liang et al. (2022), who characterized its geometric structure in CLIP. Subsequent work has sought to understand this phenomenon from multiple perspectives, including linear separability analysis (Shi et al., 2023), double-ellipsoid geometry (Levi & Gilboa, 2024), and the distinction between modality-specific and contrastive components (Fahim et al., 2024). Building on these insights, several methods exploit the gap for downstream applications such as vision model diagnosis (Zhang et al., 2023b) and cross-modal transfer via zero-centering (Zhang et al., 2024a). More recent efforts focus on mitigating the gap through learnable correction models (Park et al., 2024; Eslami & de Melo, 2024), embedding standardization (An et al., 2025), or centroid alignment for mixed-modality retrieval (Li et al., 2025). However, these methods operate in paired-data settings within a single encoder's shared space where text and modal branches are jointly trained. We demonstrate that the modality gap generalizes across six independently pretrained encoder families and can be exploited for cross-encoder modality expansion through text-only projection learning.

**LLM-Anchored Multimodal Learning.** Recent work leverages LLMs as semantic anchors for multimodal alignment, exploiting their broad semantic coverage and contextual understanding acquired through large-scale language pretraining. Generative approaches integrate LLMs with multimodal encoders for instruction tuning (Han et al., 2023) and unified cross-modal generation (Han et al., 2024). Representation-focused methods include UniBind (Lyu et al., 2024), which creates LLM-augmented unified spaces, and LLM2CLIP (Huang et al., 2024), which enhances dense caption understanding through large-scale paired training on tens of millions of image-caption pairs. More recently, E5-V (Jiang et al., 2024) and LCO-Emb (Xiao et al., 2025) show that text-only contrastive learning can enhance MLLM embedding quality without multimodal training data. However, these methods operate within unified MLLM architectures where cross-modal alignment is implicitly established during generative pretraining. In contrast, **TextME** addresses the alignment of independently trained contrastive encoders with architecturally heterogeneous embedding spaces, enabling expansion to specialized domains such as 3D, X-ray, and molecules without requiring a shared backbone.

## 6. Conclusion

We introduced **TextME**, a framework that leverages the consistent modality gap in pretrained contrastive encoders to enable text-only training for modality expansion. By projecting diverse modalities into LLM embedding space as a unified anchor, our approach preserves substantial performance of pretrained encoders across six modalities using only text descriptions for projection learning. Compared to existing paired-data methods, **TextME** reduces data requirements by over 95% while eliminating the need for paired multimodal supervision. Furthermore, the framework enables emergent cross-modal retrieval between modality pairs never seen during training (e.g., audio-to-image, 3D-to-image), demonstrating that text-anchored alignment can establish implicit correspondences across arbitrary modalities without direct cross-modal pairing. These results suggest that text-only training offers a practical pathway for integrating specialized modalities—such as medical imaging and molecular structures—into unified multimodal systems without the prohibitive cost of expert annotation.

**Limitations and Future Work.** Text-only projection training preserves categorical boundaries more effectively than fine-grained instance-level similarity, achieving 89.2% average PPR on classification but 65.3% on retrieval, as retrieval is more sensitive to distortions introduced by offset correction. Furthermore, the effectiveness of our framework depends on the geometric properties of pretrained encoders, particularly gap consistency and low-variance gap-content orthogonality. When these properties are weak, as observed with MoleculeSTM (gap consistency 0.78, orthogonality variance $\pm 0.18$), offset correction can degrade performance. While our orthogonality variance diagnostic ($r = -0.67$, $p < 0.001$) offers practitioners a principled tool for assessing encoder compatibility, developing adaptive strategies for unfavorable encoder geometry and establishing formal theoretical guarantees for the modality gap properties remain promising directions for future work.

## Acknowledgements

This work was supported by Institute of Information & communications Technology Planning & Evaluation (IITP) grant funded by the Korea government (MSIT) (IITP-2026-RS-2023-00255968, Artificial Intelligence Convergence Innovation Human Resources Development; No. RS-2022-II220680, Abductive inference framework using omni-data for understanding complex causal relations). This research was also supported by the Nano & Material Technology Development Program through the National Research Foundation of Korea (NRF) funded by Ministry of Science and ICT (RS-2024-00444182), and a grant of the Korea Health Technology R&D Project through the Korea Health Industry Development Institute (KHIDI), funded by the Ministry of Health & Welfare, Republic of Korea (Grant number: RS-2025-02310331).

## Impact Statement

This work aims to reduce the data annotation burden in multimodal learning, potentially democratizing access to multimodal AI systems in specialized domains such as medical imaging and molecular analysis where paired supervision is prohibitively expensive. While this could accelerate beneficial applications in healthcare and drug discovery, practitioners should exercise appropriate caution when deploying such models in safety-critical settings, ensuring thorough validation before clinical or real-world use. We do not foresee immediate negative societal consequences beyond those common to advances in representation learning.

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

## A. Semantic Textual Similarity Benchmark Analysis

To validate our choice of LLM embeddings as the semantic anchor space, we conduct comprehensive evaluation on the Semantic Textual Similarity (STS) benchmark suite. Table 7 presents Spearman correlation scores across six STS tasks (STS12–16 and STSBenchmark) comparing multimodal encoders (CLIP (Radford et al., 2021), LanguageBind (Zhu et al., 2023)) against LLM embedding models (NV-Embed-v2 (Lee et al., 2024), Qwen3-Embedding variants (Zhang et al., 2025)).

*Table 7.* **Semantic Textual Similarity (STS) benchmark performance.** Spearman correlation ($\rho$) scores across six STS tasks are reported, comparing multimodal encoders and LLM embedding models.

| Model | STS Tasks (Spearman $\rho$) | | | | | | Avg. |
|---|---|---|---|---|---|---|---|
| | STS12 | STS13 | STS14 | STS15 | STS16 | STSBench | |
| *Multimodal Encoders* | | | | | | | |
| CLIP | 61.87 | 63.83 | 62.09 | 76.82 | 72.89 | 72.26 | 68.29 |
| LanguageBind | 63.12 | 67.46 | 63.27 | 73.82 | 73.73 | 71.60 | 68.83 |
| *LLM Embedding Models* | | | | | | | |
| NV-Embed-v2 | 77.89 | 88.30 | 84.30 | 89.04 | 86.77 | 88.41 | 85.79 |
| Qwen3-Embed-0.6B | 79.35 | 87.31 | 79.81 | 87.28 | 87.07 | 86.51 | 84.56 |
| Qwen3-Embed-4B | 84.31 | 93.20 | 88.61 | 92.31 | 92.07 | 91.92 | 90.40 |

## B. Extended Experimental Results

This appendix provides comprehensive experimental results that supplement the main findings in Section 4. We report detailed metrics across all evaluation benchmarks to enable thorough comparison and reproducibility.

### B.1. Evaluation Benchmarks

Table 8 provides a complete overview of all evaluation benchmarks. In the main text (Table 1), we report representative benchmarks per modality for clarity: Flickr30k for image, MSVD for video, AudioCaps for audio, and DrugBank for molecule retrieval; ESC-50, ModelNet40, ScanObjectNN, and RSNA for classification.

*Table 8.* **Complete evaluation benchmarks.** All datasets used for retrieval and classification tasks are organized by task type and target modality.

| Task | Modality | Datasets |
|---|---|---|
| Text→X Retrieval | Image | COCO (Lin et al., 2014), Flickr30k (Plummer et al., 2015) |
| | Video | MSRVTT (Xu et al., 2016), MSVD (Chen & Dolan, 2011), DiDeMo (Anne Hendricks et al., 2017) |
| | Audio | AudioCaps (Kim et al., 2019), Clotho (Drossos et al., 2020) |
| | Molecule | DrugBank (Knox et al., 2024) |
| Emergent X→X | Audio→Image | FlickrNet (Senocak et al., 2018) |
| | 3D→Image | Objaverse (Deitke et al., 2023) |
| Zero-shot Cls. | 3D | ModelNet40 (Sun et al., 2022), ScanObjectNN (Uy et al., 2019) |
| | Audio | AudioSet (Gemmeke et al., 2017), ESC-50 (Piczak, 2015) |
| | X-ray | RSNA (RSNA, 2018) |

### B.2. Detailed Cross-Modal Retrieval Performance

Table 1 in the main text reports representative R@1 metrics for brevity. Here, we provide complete retrieval results including R@5 metrics, which capture whether relevant items appear within the top-5 retrieved candidates. This relaxed criterion is particularly informative for assessing approximate semantic alignment quality.

As shown in Table 9, **TextME** consistently achieves higher PPR on R@5 compared to R@1 across all benchmarks (e.g., 75.6% vs. 59.3% on COCO, 98.4% vs. 89.7% on MSVD). This pattern indicates that text-only training effectively preserves coarse-grained semantic structure, with the performance gap primarily arising from fine-grained ranking precision. Notably, on MSVD, ours achieves near-perfect R@5 preservation (98.4%), suggesting that the learned projections successfully capture the underlying semantic relationships for video-text alignment.

*Table 9.* **Detailed Text→Image and Text→Video retrieval performance.** R@1 and R@5 metrics are reported. PPR denotes Performance Preservation Ratio (%) relative to pretrained encoders (CLIP for Image, ViCLIP for Video).

| | Image Retrieval | | | | Video Retrieval | | | |
|---|---|---|---|---|---|---|---|---|
| | COCO | | Flickr30k | | MSRVTT | | MSVD | |
| | R@1 | R@5 | R@1 | R@5 | R@1 | R@5 | R@1 | R@5 |
| Pretrained | 48.29 | 72.51 | 77.70 | 94.16 | 37.00 | 63.70 | 51.06 | 78.29 |
| **TextME** | 28.63 | 54.81 | 51.66 | 77.90 | 26.40 | 50.50 | 45.82 | 77.01 |
| PPR (%) | 59.3 | 75.6 | 66.5 | 82.7 | 71.4 | 79.3 | 89.7 | 98.4 |

*Table 10.* **Detailed Text→Audio and Text→Molecule retrieval performance.** R@1 and R@5 metrics are reported for audio, and MRR@10 and MRR@20 for molecule retrieval. PPR denotes Performance Preservation Ratio (%) relative to pretrained encoders (CLAP for Audio, MoleculeSTM for Molecule).

| | Audio Retrieval | | | | Molecule Retrieval | |
|---|---|---|---|---|---|---|
| | AudioCaps | | Clotho | | DrugBank | |
| | R@1 | R@5 | R@1 | R@5 | MRR@10 | MRR@20 |
| Pretrained | 22.47 | 54.43 | 16.90 | 39.75 | 79.19 | 69.17 |
| **TextME** | 15.35 | 43.88 | 7.81 | 23.81 | 34.75 | 27.97 |
| PPR (%) | 68.3 | 80.6 | 46.2 | 59.9 | 43.9 | 40.4 |

The audio retrieval results exhibit a similar trend, with R@5 preservation consistently exceeding R@1. However, the molecule retrieval task shows lower overall preservation rates, which we attribute to the highly specialized vocabulary in chemical descriptions that differs substantially from the general text distributions used in LLM pretraining.

### B.3. Detailed Zero-shot Classification Results

Table 11 provides complete zero-shot classification results including Top-5 accuracy.

*Table 11.* **Detailed zero-shot classification performance.** Top-1 and Top-5 accuracy are reported across audio (ESC-50) and 3D (ModelNet40, ScanObjectNN) modalities, along with mAP for AudioSet. PPR denotes Performance Preservation Ratio (%) relative to pretrained encoders.

| Method | AudioSet mAP | ESC-50 Top-1 | Top-5 | ModelNet40 Top-1 | Top-5 | ScanObjectNN Top-1 | Top-5 |
|---|---|---|---|---|---|---|---|
| Pretrained | 9.32 | 85.20 | 97.70 | 67.75 | 90.07 | 42.21 | 77.23 |
| LanguageBind | 18.33 | 94.00 | 99.70 | – | – | – | – |
| Ex-MCR | 6.67 | 71.20 | 96.80 | 66.53 | 93.60 | 40.31 | 77.20 |
| Naïve | 1.14 | 2.90 | 8.45 | 0.81 | 8.95 | 3.32 | 30.52 |
| COX[†] | 1.26 | 2.00 | 10.00 | 4.05 | 13.70 | 2.84 | 26.68 |
| **TextME** | 5.80 | 77.25 | 96.85 | 70.86 | 92.14 | 42.15 | 77.89 |
| PPR (%) | 62.2 | 90.7 | 99.1 | 104.6 | 102.3 | 99.9 | 100.9 |

Notably, **TextME** achieves PPR exceeding 100% on 3D classification benchmarks (ModelNet40: 104.6%, ScanObjectNN: 99.9%), demonstrating that text-only training can sometimes improve upon pretrained encoder performance. Top-5 preservation consistently exceeds Top-1, indicating that approximate categorical boundaries are well-preserved even when precise rankings differ.

### B.4. Alternative Encoder Family

To verify that **TextME** generalizes beyond the default CLIP ViT-H-14 used in the main experiments, we evaluate the framework with EVA-02-CLIP-E/14+ (Sun et al., 2023), which incorporates masked image modeling during pretraining and thus represents a distinct encoder family. Table 12 reports retrieval performance on Flickr30k and COCO.

**TextME** successfully transfers to EVA-CLIP, achieving 51–81% PPR across benchmarks without any paired supervision

*Table 12.* **Alternative encoder family evaluation. TextME** is applied to EVA-02-CLIP-E/14+ and compared against the default CLIP ViT-H-14. PPR denotes Performance Preservation Ratio (%) relative to each encoder's pretrained performance.

| | Flickr30k | | | COCO | | |
|---|---|---|---|---|---|---|
| Model | R@1 | R@5 | R@10 | R@1 | R@5 | R@10 |
| *CLIP ViT-H-14* | | | | | | |
| Pretrained | 77.7 | 94.2 | 96.6 | 48.3 | 73.4 | 82.9 |
| **TextME** | 51.7 | 77.9 | 86.0 | 28.6 | 54.6 | 66.1 |
| PPR (%) | 66.5 | 82.7 | 89.0 | 59.2 | 74.4 | 79.7 |
| *EVA-02-CLIP-E/14+* | | | | | | |
| Pretrained | 76.1 | 92.6 | 96.1 | 46.2 | 70.5 | 79.3 |
| **TextME** | 39.1 | 68.5 | 78.1 | 20.6 | 43.3 | 54.3 |
| PPR (%) | 51.4 | 74.0 | 81.3 | 44.6 | 61.4 | 68.5 |

and significantly outperforming all unpaired baselines (Table 1). The lower PPR compared to CLIP ViT-H-14 can be attributed to differences in geometric properties. Table 13 reports the four properties measured in Section 4.3 for both encoders. EVA-CLIP exhibits weaker satisfaction of properties (i) and (ii), indicating that its modality gap is less amenable to characterization by a single global offset vector.

*Table 13.* **Geometric properties of CLIP and EVA-CLIP.** †: weaker satisfaction ($>0.1$ for (i), $<0.96$ for (ii)).

| Encoder | Mod. | (i)$\downarrow$ | (ii)$\uparrow$ | (iii)$\downarrow$ | (iv)$\downarrow$ |
|---|---|---|---|---|---|
| CLIP ViT-H-14 | Image | $.28^{\dagger}\pm.11$ | $.97\pm.00$ | $.00\pm.00$ | $.00\pm.11$ |
| EVA-02-CLIP-E/14+ | Image | $.37^{\dagger}\pm.07$ | $.95^{\dagger}\pm.01$ | $-.01\pm.09$ | $.00\pm.07$ |

## C. Implementation Details

### C.1. Model Architecture

Each projection network $P_m$ is implemented as a 2-layer MLP with GeLU activation as

$$P_m(x) = W_2 \cdot \text{GeLU}(W_1 \cdot x + b_1) + b_2 \tag{6}$$

where the hidden dimension matches the source encoder's embedding dimension $d_m$, and the output dimension is fixed at $d_h = 2560$ to match Qwen3-Embedding. Each projection network contains approximately 10M trainable parameters.

### C.2. Training Configuration

The hyperparameter configuration for training is summarized in Table 14. All projection networks are trained with the same configuration across modalities.

*Table 14.* **Training hyperparameters.** All projection networks are trained with the same configuration across modalities.

| Hyperparameter | Value |
|---|---|
| Batch size | 512 |
| Optimizer | AdamW ($\beta_1 = 0.9$, $\beta_2 = 0.999$) |
| Weight decay | 0.01 |
| Learning rate | $5 \times 10^{-4}$ |
| LR schedule | Cosine annealing |
| Training epochs | 50 |
| Temperature $\tau$ | 0.07 |
| Hard negative range | $[0.1 \cdot s_i, 0.9 \cdot s_i]$ |
| Precision | fp16 |

## C.3. Offset Computation

For each modality, we compute centroids using 5,000 randomly sampled text-modal pairs. Table 15 summarizes the datasets used for offset estimation across all evaluated encoders.

*Table 15.* **Datasets used for offset computation.** For each encoder, centroids are estimated using 5,000 randomly sampled text-modal pairs from the listed datasets.

| Encoder | Modality | Offset Dataset | Domain |
|---|---|---|---|
| CLIP | Image | COCO (Lin et al., 2014) | Natural images |
| ViCLIP | Video | MSRVTT (Xu et al., 2016) | Web videos |
| CLAP | Audio | AudioCaps (Kim et al., 2019) | Audio events |
| Uni3D | 3D | Objaverse (Deitke et al., 2023) | Synthetic objects |
| CXR-CLIP | X-ray | CheXpert (Irvin et al., 2019) | Medical imaging |
| MoleculeSTM | Molecule | PubChem (Kim et al., 2025) | Chemical compounds |
| LanguageBind | Text | COCO (Lin et al., 2014) | Natural images |

Text inputs are tokenized with a maximum sequence length of 77 tokens. Offsets are pre-computed once and remain fixed throughout training.

We note that the choice of dataset for offset computation may influence both the estimated gap properties and downstream performance. Since the modality gap is computed as the difference between text and modal centroids, the semantic distribution of the offset dataset could affect the resulting offset vector. For instance, encoders whose offset datasets closely match the evaluation domain may exhibit more favorable gap properties, while domain mismatch between offset computation and downstream evaluation could introduce additional variability. Although our experiments demonstrate robust performance across diverse benchmarks, a systematic investigation of how offset dataset selection affects alignment quality remains an important direction for future work.

## C.4. Computational Resources

All experiments are conducted on a single NVIDIA A6000 GPU (48GB). Training time averages 2 hours per modality with peak memory usage of approximately 8GB.

# D. COX Baseline Implementation

Since the original COX (Huang et al., 2025) codebase is not publicly available, we re-implemented the method following the paper specifications with adaptations for our zero-shot evaluation setting.

**Architecture.** We employ Vision Transformer Tiny (ViT-T/16) as the encoder backbone with 12 layers, 3 attention heads, and embedding dimension 192. Following the original design, we incorporate a Variational Information Bottleneck (VIB) layer with stochastic dimensionality reduction to 256 dimensions.

**Training Protocol.** We follow the two-stage methodology: (1) supervised pre-training on labeled target data for 10 epochs, and (2) information bottleneck fine-tuning for 50 epochs. We use batch size 256, Adam optimizer with learning rate $1 \times 10^{-3}$ and weight decay $1 \times 10^{-5}$.

**Key Difference from TextME.** COX requires labeled target modality data ($\sim$10K samples), trains encoders from scratch ($>$300M parameters), and demands architectural alignment between source and target encoders. In contrast, **TextME** leverages pre-trained encoders with only text descriptions, requires merely $\sim$10M trainable parameters, and imposes no architectural constraints.

# E. Additional Ablation Studies

## E.1. Sample Size for Offset Estimation

We investigate sensitivity to the number of samples used for computing centering offsets.

Results demonstrate that offset estimation is robust to sample size, with performance plateauing between 1,000–10,000 samples. Even with only 100 samples, the method achieves 90% of default performance, validating the efficiency of our

*Table 16.* **Impact of sample size for offset estimation.** Performance remains stable for $N \geq 1{,}000$.

| # Samples | Audio R@1 | 3D Top-1 | Mol. MRR | Rel. Perf. |
|---|---|---|---|---|
| 100 | 14.91 | 70.66 | 34.75 | 90% |
| 500 | 14.77 | 70.58 | 33.05 | 95% |
| 1,000 | 14.89 | 70.62 | 36.44 | 97% |
| 5,000 (default) | 15.35 | 70.86 | 34.75 | 100% |
| 10,000 | 14.95 | 70.58 | 32.20 | 100% |

approach.

### E.2. Offset Noise Sensitivity

To assess robustness to offset estimation errors, we perturb the pre-computed offset $\Delta$ with additive Gaussian noise: $\Delta' = \Delta + \mathcal{N}(0, \sigma^2 I)$.

*Table 17.* **Offset noise sensitivity.** Text→Audio retrieval (R@1), 3D classification (Top-1), and Text→Molecule retrieval (MRR) are reported. Performance degrades gracefully for $\sigma < 0.05$.

| Noise $\sigma$ | Audio R@1 | 3D Top-1 | Mol. MRR |
|---|---|---|---|
| 0.000 | 14.95 | 70.46 | 27.97 |
| 0.001 | 14.95 | 70.30 | 24.58 |
| 0.01 | 15.04 | 67.50 | 22.88 |
| 0.05 | 14.93 | 34.32 | 17.80 |
| 0.10 | 14.25 | 14.91 | 11.02 |

Audio demonstrates remarkable stability, maintaining near-baseline performance even at $\sigma = 0.10$. In contrast, 3D and Molecule show sharper degradation at $\sigma \geq 0.05$, indicating these modalities require more precise offset estimation. For practical deployment, 5,000 samples provide sufficient precision with empirical standard error well below $\sigma = 0.01$.

### E.3. Text Encoder Ablation

In the main experiments, we adopt LanguageBind (Zhu et al., 2023) as the shared text encoder for all modalities. To verify that **TextME** is not dependent on a specific text encoder, we replace it with CLIP ViT-H-14[1] and compare performance across retrieval and classification tasks. Table 18 reports the results.

*Table 18.* **Text encoder ablation.** Performance comparison between LanguageBind and CLIP ViT-H-14 as the shared text encoder across retrieval (AudioCaps R@1, DrugBank MRR) and classification (ESC-50, ModelNet40, RSNA Top-1 accuracy) tasks. Both encoders produce competitive results without any modification to the framework.

| Text Encoder | Retrieval | | Classification | | |
|---|---|---|---|---|---|
| | ACaps R@1 | Drug. MRR | ESC Top-1 | MN40 Top-1 | RSNA Top-1 |
| LanguageBind (768d) | 14.5 | 44.1 | 74.3 | 98.8 | 45.0 |
| CLIP ViT-H-14 (1024d) | 15.9 | 51.7 | 86.7 | 97.9 | 48.3 |

CLIP ViT-H-14 achieves stronger performance on most benchmarks, which we attribute to its larger embedding dimensionality (1024 vs. 768) and broader pretraining coverage. Importantly, **TextME** operates with both encoders without any architectural modification, confirming that the framework generalizes across text encoder families. We adopt LanguageBind as the default in the main experiments because it is already fine-tuned for multimodal expansion across five modalities, and using a single shared text encoder enables cross-modal retrieval between arbitrary modality pairs.

---

[1] `laion/CLIP-ViT-H-14-laion2B-s32B-b79K` (Cherti et al., 2023)

## E.4. LLM-Synthesized Training Descriptions

To evaluate whether LLM-generated descriptions can substitute for manually curated domain-specific text, we train projection networks using synthetic captions. Specifically, we prompt Qwen3-14B (Yang et al., 2025)[2] with 4-shot in-context examples to rewrite image captions from ConceptualCaptions (Changpinyo et al., 2021) into audio event descriptions, yielding approximately 332K synthetic captions from which we sample 100K for projection training. Table 19 compares three training data sources on audio benchmarks under identical hyperparameters.

*Table 19.* **Effect of LLM-synthesized training descriptions.** Synthetic captions generated by Qwen3-14B achieve competitive performance relative to domain-specific captions, substantially outperforming general-purpose text.

| Training Data | AudioCaps R@1 | ESC-50 Top-1 |
|---|---|---|
| Domain captions | **14.5** | 74.3 |
| Synthesized (Qwen3-14B) | 12.5 | **79.2** |
| Wiki1M (general) | 7.3 | 62.0 |

LLM-synthesized descriptions achieve 86% of the domain-specific retrieval performance on AudioCaps while surpassing it on ESC-50 classification by 4.9 points, which we attribute to the broader sound vocabulary coverage in the synthetic corpus. Both substantially outperform general-purpose text (Wiki1M), confirming that domain relevance remains important for projection training. These results suggest that when curated domain text is unavailable, a single LLM prompting session can produce a viable training corpus without manual annotation or target modality data acquisition.

## E.5. Fallback Strategy for Weak Encoder Geometry

We investigate whether more expressive offset correction strategies can improve upon the single global offset for modalities with unfavorable geometric properties. We focus on Molecule and Audio, which exhibit the highest orthogonality variance (Table 2). For local structure, multi-cluster offset ($K = 7$) partitions the 5K offset samples via $k$-means and applies a separate offset per cluster at inference. For non-linearity, a learned 2-layer MLP trained on 5K paired samples replaces the fixed offset with a nonlinear mapping. Table 20 reports the results.

*Table 20.* **Fallback strategy ablation.** Neither multi-cluster nor learned MLP offset outperforms the single global offset, suggesting that capturing finer-grained gap structure from limited unpaired data remains an open challenge.

| Method | Mol. MRR | Audio R@1 |
|---|---|---|
| No offset | 36.4 | 8.7 |
| Global offset ($K$=1, unpaired) | 34.8 | **15.4** |
| Multi-cluster offset ($K$=7, unpaired) | 34.8 | 12.5 |
| Learned MLP offset (paired) | 33.1 | 15.3 |

Neither alternative outperforms the global offset. The multi-cluster approach degrades Audio retrieval, because partitioning 5K samples into 7 clusters yields insufficient per-cluster coverage for reliable centroid estimation. The learned MLP, despite using paired supervision, shows no improvement, indicating that the residual gap structure beyond the global offset is too subtle to capture from limited samples. These results validate the use of a single global offset as the default strategy while highlighting that developing more effective correction mechanisms for encoders with weak geometric properties remains a promising direction for future work.

# F. Embedding Visualizations

Figure 7 extends the t-SNE analysis from Figure 5 to the remaining four modalities. Image (CLIP) and Audio (CLAP) both show progressive reduction in modality separation across the three stages, with text and modal distributions increasingly overlapping after projection. 3D (Uni3D) exhibits the clearest transition, moving from fully separated clusters to a shared region after projection, consistent with its high CKA (0.904) and downstream PPR. X-ray (CXR-CLIP) presents a distinct pattern: text embeddings form a small isolated cluster that remains separated after centering but achieves partial mixing after projection, consistent with its low CKA (0.472) despite high classification PPR (88.5%).

---

[2] `Qwen/Qwen3-14B`

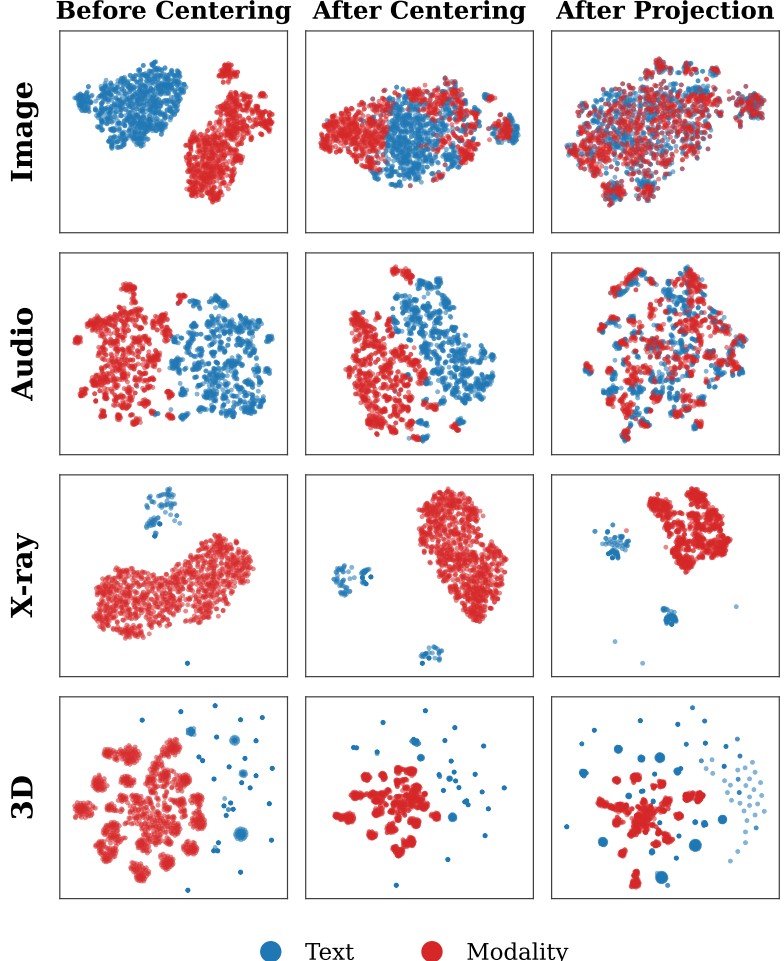

*Figure 7.* **t-SNE visualization for Image, Audio, X-ray, and 3D modalities.** Text (blue) and modal (red) embeddings are shown before centering, after centering, and after projection. Image, Audio, and 3D show progressive overlap across stages, while X-ray achieves partial alignment primarily through projection.

## G. Theoretical Interpretation

The empirical geometric analysis in Section 4.3 can be grounded in the theoretical framework of Zhang et al. (2024a). Their Proposition 1 decomposes the difference between paired embeddings as $e_x - e_t = c_\perp + \epsilon$, where $c_\perp$ is a constant gap vector orthogonal to the embedding span and $\epsilon \sim \mathcal{N}(0, \sigma^2 I)$ is alignment noise. Their Lemma 1 further proves that contrastive gradients lie entirely within the embedding span, meaning the dimensions where the gap resides receive no gradient update during optimization. This provides the theoretical basis for why centering can remove $c_\perp$ while preserving semantic content, with the residual correction reducing to $\epsilon$.

The four geometric properties measured in Section 4.3 serve as empirical diagnostics for how well a given encoder satisfies the assumptions underlying this decomposition. Properties (i) and (ii) assess whether a single constant $c_\perp$ adequately characterizes the modality gap, while properties (iii) and (iv) assess whether the residual $\epsilon$ remains small enough for offset correction to preserve semantic relationships. When these properties are well-satisfied, the theoretical assumptions hold closely, leading to higher downstream performance. This connection accounts for why TextME achieves strong PPR for encoders such as Uni3D (orthogonality variance $\pm 0.04$) but degrades for MoleculeSTM ($\pm 0.18$), where the residual $\epsilon$ is large and the constant-gap assumption breaks down. Formally extending these guarantees to modality expansion without paired supervision remains an open theoretical question that we plan to explore in future work.

