# OpenReview forum: "TextME: Bridging Unseen Modalities Through Text Descriptions"
_ICML.cc/2026/Conference — ICML 2026 regular_

### Official Review · Reviewer_pvs8 · 2026-03-10

**Soundness:** 2
**Presentation:** 3
**Significance:** 2
**Originality:** 2
**Overall Recommendation:** 3
**Confidence:** 3

**Summary:**

This paper proposes TextME, a text-only modality expansion framework that projects diverse modalities into large language model embedding space without relying on large-scale paired datasets. Leveraging the geometric structure of pretrained contrastive encoders, TextME enables zero-shot cross-modal transfer using only text descriptions. Empirical results across multiple domains—including image, video, audio, 3D, X-ray, and molecular data—demonstrate that text-only supervision can effectively preserve pretrained encoder performance and support emergent cross-modal retrieval between previously unaligned modality pairs. The findings highlight text-only training as a practical and scalable alternative to paired supervision for modality expansion.

**Compliance With Llm Reviewing Policy:**

Affirmed.

**Final Justification:**

Although the authors clarify that the suggested scaling-up experiment deviates from the core objective and target scenarios, this makes the proposed method less solid. Besides, while the performance drop due to the use of general-purpose text is a natural phenomenon, it also reveals a limitation of the method: it only works when using task-related textual data. In summary, I tend to keep my original score. Considering that the other reviewers gave positive scores, I have moderately lowered my confidence.

**Key Questions For Authors:**

- The experiments in this paper are conducted only on small-scale datasets. My primary concern is whether the proposed method can scale effectively to larger datasets.
- As shown in Table 5, training with general-purpose text corpora significantly reduces downstream performance. Therefore, the proposed method still requires large amounts of text descriptions paired with the target modality for projection network training.
- Why is the zero-shot performance on classification tasks in Table 1 significantly higher than that on retrieval tasks?
- Table 4 shows significant differences among various anchor spaces across downstream tasks; however, the authors do not provide convincing analyses. For example, it remains unclear why large-scale language pretraining outperforms large-scale vision-language contrastive training on AudioCaps.

**Limitations:**

yes

**Strengths And Weaknesses:**

### Strengths
1. The research motivation of this paper is well justified.
2. This paper is well written.

### Weaknesses
1. The analysis in this paper is not sufficiently in-depth, and the experiments are not comprehensive enough.

---

> ### Author Rebuttal · Authors · 2026-03-31
>
> We thank the reviewer for raising important questions that helped us strengthen the paper. We address each below with new experiments.
>
> ---
> ## Q1. Scalability to larger datasets
>
> As suggested, we scaled the 3D modality expansion experiment to 1M text descriptions (10x over our default) from Objaverse (Deitke et al., 2023) as it provides the largest publicly available collection of modality-specific text among our evaluated domains.
>
> | Training Size | ModelNet40 | ScanObjectNN |
> |:--|---|---|
> | 100K (default) | **70.86** | 42.15 |
> | 1M (10x) | 70.60 | **45.30** |
>
> This scaling yielded a clear improvement on ScanObjectNN, the more challenging benchmark with larger distribution shift; showing that TextME’s generalization scales effectively with more data. The results also suggest that domain relevance matters more than raw scale. Switching from general to domain-specific text at 100K yields a $6\times$ improvement (Table 5), far exceeding the gain from $10\times$ more data. This is a practical advantage of TextME, as it achieves strong performance with modest data requirements rather than relying merely on scale.
>
> ---
> ## Q2. Performance drop with general-purpose text and the need for paired data
>
> We first note that TextME requires only unpaired text descriptions, not (modality, text) pairs. For example, it uses audio event captions but never actual audio files. Moreover, such domain-relevant text already exists at scale in public repositories, e.g., PubChem with 119M compound descriptions [1], Objaverse with 800K+ annotated 3D captions [2], and MIMIC-CXR with 227K radiology reports [3]. 100K samples TextME demands thus represents only a small fraction of these resources, requiring no additional collection effort.
>
> Second, when existing domain text is unavailable, LLM-synthesized descriptions provide a practical alternative requiring only API calls with no manual annotation or data acquisition.
>
> | Training Text Data | AudioCaps R@1 | ESC-50 Acc |
> |:--|---|---|
> | AudioCaps captions (domain-specific) | 14.5 | 74.3 |
> | Synthesized captions (LLM-gen.) | 12.5 | 79.2 |
> | Wiki1M (general text) | 7.3 | 62.0 |
>
> As shown in the table, TextME trained with synthetic captions achieved 86% of the original’s performance in AudioCaps retrieval, while even exceeding it in ESC-50 classification.
>
>
> [1] Kim et al., Nucleic Acids Research, 2025.
> [2] Deitke et al., CVPR, 2023.
> [3] Johnson et al., Scientific Data, 2019.
>
>
> ---
> ## Q3. Classification vs. retrieval performance gap
>
> We attribute this to task granularity, as discussed in Section 4.2 (lines 296-302). Classification needs only category boundaries, while retrieval requires instance-level ranking sensitive to offset distortion, as even small perturbations can alter the related ordering of candidates. Such a difference leads to a significant gap in performance preservation ratio (PPR): 104.6% on ModelNet40 vs. 43.9% on DrugBank. This is further supported by R@5 consistently showing higher PPR than R@1 (e.g., MSVD 89.7% to 98.4% in Appendix Table 8), confirming that coarse semantic structure is well preserved and the gap primarily arises from fine-grained ranking precision. We discuss a practical mitigation via multi-anchor ensemble in Q4 below.
>
> ---
> ## Q4. Anchor space analysis
>
> We attribute the performance variation across anchors to differences in their training objectives (Section 3.3). LLM embeddings achieve higher STS performance (Spearman $\rho = $85 ~ 90, Appendix Table 6) than multimodal encoders ($\rho = $67 ~ 68), suggesting that LLM embeddings capture fine-grained semantic similarity more effectively, which explains their stronger performance on retrieval tasks such as AudioCaps. In contrast, classification requires relatively coarse decision boundaries, which is better suited to multimodal encoders pretrained with cross-modal contrastive objective rather than text-text similarity. An exception is the molecule, which shows unstable performance regardless of anchor choice; we attribute this to the high gap orthogonality variance (property (iv), Table 2) rather than anchor space itself.
>
> To practically address this instability across anchors, we explored multi-anchor ensemble via score-level late fusion, which achieves consistently strong performance across all tasks.
>
> | **Anchor** | **AudioCaps R@1** | **DrugBank Acc@10** | **MN40 Acc** | **ESC-50 Acc** | **RSNA Acc** |
> |:--|---|---|---|---|---|
> | CLIP (1024D) | 15.9 | 36.4 | 78.0 | **86.7** | 48.3 |
> | LanguageBind (768D) | 14.5 | 29.7 | **81.1** | 74.7 | 45.0 |
> | NV-Embed-v2 (4096D) | 16.2 | 26.3 | 78.3 | 79.4 | **48.6** |
> | Qwen3-Embed (2560D) | 15.4 | 34.8 | 79.0 | 77.2 | 46.6 |
> | **Ensemble** | **19.2** | **41.5** | 79.7 | **86.7** | 46.9 |

---

> > ### Author Rebuttal · Reviewer_pvs8 · 2026-04-03
> >
> > The additional experimental results provided by the authors show that after expanding the data by 10 times, ModelNet40 does not improve as significantly as ScanObjectNN, indicating that the scaling capability of this method is limited.
> >
> > Moreover, although the authors emphasize that TextME requires only unpaired text descriptions, not (modality, text) pairs, the use of general-purpose text leads to poor performance, so the method proposed in this paper still has obvious limitations.
> >
> > In summary, I tend to maintain the current score.

---

> > > ### Author Response · Authors · 2026-04-07
> > >
> > > Thank you for the continued engagement. We address the two remaining concerns below.
> > >
> > > ---
> > > ## On Scaling Capability
> > >
> > > We would first like to clarify that the suggested scaling-up experiment deviates from our core objective and target scenarios. The primary motivation of our work is the practical difficulty of obtaining modality-paired supervision. This challenge arises not only from the inherent difficulty of pairing different modalities, but also from the scarcity of available data due to privacy concerns and the demand of expert knowledge (lines 33-37).
> > >
> > > To address this, we developed TextME that is specifically designed for modality expansion using **limited amounts** of text-only data. TextME already achieved significant performance with only 100K unpaired text descriptions, e.g., notably outperforming pretrained Uni3D and Ex-MCR using paired data on ModelNet (Table 1 of the main paper). Given this, the additional performance gains of scaling to 1M descriptions rather underscore the intended data-efficiency of TextME. We thus respectfully argue that these results directly align with our stated objective of achieving high utility under data-constrained conditions.
> > >
> > >
> > > ---
> > > ## On Performance with General-Purpose Text
> > >
> > > Performance drop due to the use of general-purpose text is a natural phenomenon. Recent advances in mechanistic interpretability suggest that the latent representation space of LLMs can be viewed as a collection of low-dimensional manifolds, each partitioned by specific domains [1,2]. For example, knowledge regarding specialized domains, such as molecular structures, likely resides within a distinct manifold that is geometrically distinct from that of general-purpose text. Consequently, projecting special domain data like molecules into the LLM’s anchor space requires domain-matched textual descriptions that are geometrically proximal to the relevant knowledge clusters in the space. Using general-purpose text for such alignment is inherently limited because it cannot accurately capture or navigate toward these specialized manifolds.
> > >
> > > [1] Anthropic, Scaling Monosemanticity: Extracting Interpretable Features from Claude 3 Sonnet, 2024
> > >
> > > [2] Saglam et al., Large Language Models Encode Semantics and Alignment in Linearly Separable Representation, IJCNLP 2025
> > >
> > >
> > > ---
> > >
> > > We hope these clarifications address the reviewer’s remaining concerns and demonstrate that TextME achieves its ultimate goal within the scope of our contribution.

---

### Official Review · Reviewer_E4nS · 2026-03-11

**Soundness:** 3
**Presentation:** 3
**Significance:** 3
**Originality:** 3
**Overall Recommendation:** 5
**Confidence:** 4

**Summary:**

This paper focuses on the problem where expanding to new modalities usually depends on large paired datasets. These data are expensive or infeasible in domains. The authors propose a text-only modality expansion framework that leverages the geometric “modality gap” in pretrained contrastive encoders. The core idea is to estimate modality-specific text/modal centroids to form an interchangeable space, and then train lightweight projectors using only text descriptions to map each modality into a shared LLM embedding space. At inference time, centered modality embeddings are passed through the same projector, enabling zero-shot transfer without paired multimodal supervision. The paper evaluates the method across six modalities (image, video, audio, 3D, X-ray, and molecules) and shows competitive performance relative to paired-data methods, while requiring only text data for training and enabling emergent retrieval between unseen modality pairs.

**Compliance With Llm Reviewing Policy:**

Affirmed.

**Final Justification:**

Most of my concerns have been addressed. Although additional experiments are needed to further justify the approach, the current results are novel and quite promising. Therefore, I am increasing my score to 5.

**Key Questions For Authors:**

- How sensitive is the method to the choice of pretrained encoders? Do the authors have evidence that the centering-based alignment works consistently across different encoder families (i.e., dino for image), or is it specific to certain pretrained models?
- Why is a single global centroid/offset sufficient? Have the authors explored more expressive alternatives, such as class-conditional, cluster-wise, or local alignment strategies?

**Limitations:**

yes

**Strengths And Weaknesses:**

Strengths:
+ The paper targets a bottleneck in multimodal learning: dependence on expensive paired supervision. It formulates a practically meaningful alternative based on modality-only training.
+ The use of modality-gap centering to create an interchangeable space is simple, intuitive, and well aligned with the proposed training/inference pipeline. The method reuses pretrained encoders and only trains lightweight projectors, which makes the approach easy to understand and potentially easy to adopt.
+ The evaluation spans six very different modalities and includes text-to-X retrieval, zero-shot classification, and emergent X-to-X retrieval. This breadth strengthens the claim that the framework is not limited to a narrow setting.
+ Compared with paired-data baselines requiring 1M–10M pairs, TextME uses 100K text descriptions and still preserves a substantial portion of pretrained performance, including very strong zero-shot classification results and nontrivial emergent cross-modal transfer.

Weaknesses:
- The term “text-only” is somewhat misleading. More precisely, the method avoids the need for large-scale paired supervision, rather than eliminating the need for modality data altogether. It still depends on pretrained modality encoders and therefore inherits substantial information from existing modality-specific models.
- The method relies heavily on the geometric properties of pretrained encoders. Its effectiveness appears to depend on the presence of a favorable modality-gap structure in the embedding space. As a result, the approach may not generalize uniformly across encoder families or settings where such geometry is weaker or less well aligned.
- While this design is simple and efficient, a single global correction may be insufficient to capture more fine-grained or local structure across modalities. This limitation is also consistent with the weaker performance on more challenging retrieval settings.
- The paper does not directly quantify representation loss after centering and projection. Although downstream preservation is evaluated through PPR, the paper does not explicitly measure how much semantic structure or instance-level information is lost during the transformation into the shared space.
- Minor weakness:
  - The main practical value lies in lowering the cost of multimodal expansion, rather than achieving clear gains over existing multimodal systems. As a result, the broader engineering impact may be somewhat limited.
  - The current explanation is suggestive and insightful, but it still reads more as an empirical observation supported by geometric analysis than as a rigorous theoretical account with strong guarantees.

---

> ### Author Rebuttal · Authors · 2026-03-31
>
> We sincerely thank the reviewer for the careful and insightful feedback.
>
> ---
> ## W1. The term "text-only" is misleading
>
> We appreciate this clarification. "Text-only" refers to the projection training stage, where no paired (text, modality) data is used. Pretrained encoders are reused frozen, the same assumption as Ex-MCR and LanguageBind. Offset computation uses 5K unlabeled modal samples without text pairing. We will revise the terminology to "text-only projection learning" in the revision.
>
> ---
> ## W2. Dependence on geometric properties
>
> We agree that encoder geometry is a key factor governing TextME's effectiveness. However, we would emphasize that leveraging these geometric properties to enable text-only projection learning with a single global offset is precisely our contribution. Furthermore, prior work has not characterized when text-only expansion succeeds or fails across diverse encoders. Our analysis across 6 encoder families establishes orthogonality variance as a concrete predictive tool ($r$ = -0.67, $p$ < 0.001), enabling practitioners to assess the geometric compatibility of candidate encoders prior to training. See our response to W3 & Q2 below for experiments on more expressive offset alternatives.
>
> ---
> ## W2. Dependence on geometric properties & W3/Q2. Single global correction may be insufficient
>
> We agree that encoder geometry is a key factor governing TextME's effectiveness. However, we would emphasize that leveraging these geometric properties to enable text-only projection learning with a single global offset is precisely our contribution. Prior work has not characterized when text-only expansion succeeds or fails across diverse encoders. Our analysis across 6 encoder families establishes orthogonality variance as a concrete predictive tool ($r$ = -0.67, $p$ < 0.001), enabling practitioners to assess the geometric compatibility of candidate encoders prior to training.
>
> Regarding whether more expressive corrections (e.g., cluster-wise or learned offsets) could improve over the single global offset, we tested multi-cluster offset (K=7) with 5K samples via k-means and a for non-linearity, a learned 2-layer MLP trained on 5K paired samples replaces the fixed offset with a nonlinear mapping. Please see our response to Reviewer isru (Q3) for the full experimental results and discussion.
>
> ---
> ## W4. Representation loss quantification
>
> We quantified semantic structure preservation at each stage (original → centered → projected) using three metrics: **Matched CosSim** (similarity between paired text-modal embeddings, higher is better), **Unmatched CosSim** (similarity between unpaired embeddings, lower is better), and **Gap** (difference between the two, higher means better discrimination). We also report **CKA** between projected and original embeddings to measure overall structural preservation.
>
> | Modality | **Gap (orig → proj)** | **CKA** |
> |:--|:--:|:--:|
> | Audio (CLAP) | 0.429 → **0.524** (+22\%) | 0.799 |
> | 3D (Uni3D) | 0.133 → **0.182** (+37\%) | **0.941** |
> | Image (CLIP) | 0.299 → **0.438** (+46\%) | 0.846 |
> | X-ray | 0.126 → **0.282** (+124\%) | 0.566 |
> | Molecule | 0.690 → **0.511** (-26\%) | 0.723 |
>
> Gap improves through the framework for all modalities except Molecule, confirming that centering and projection enhance discriminative structure. CKA reveals that modalities with strong geometric properties retain high structural similarity (3D: 0.94, Image: 0.85, Audio: 0.80), while weaker properties lead to more distortion (X-ray: 0.57, Molecule: 0.72), consistent with downstream PPR.
>
> ---
> ## Minor weakness: Practical value and theoretical guarantees
>
> We respectfully note that reducing data requirements by over 100$\times$ (from 1-10M paired samples to 100K unpaired text) while retaining $\sim$ 88\% PPR enables modality expansion in domains where paired annotation is prohibitively expensive, such as medical imaging and molecular analysis. We view this efficiency gain not as a trade-off against performance, but as enabling modality expansion in settings where prior methods are simply inapplicable.
>
> Regarding theoretical guarantees, our empirical analysis across 6 encoders with the orthogonality variance diagnostic ($r$ = -0.67, $p$ < 0.001) provides principled guidance. We agree that deriving formal bounds would further strengthen the framework, and consider this a promising direction for future work.
>
> ---
> ## Q1. Sensitivity to encoder choice
>
> Our experiments span 6 diverse encoder families. We are running additional experiments with EVA-CLIP [1], which shares the same contrastive objective as CLIP but uses a different architecture to test whether TextME's alignment transfers across encoder families. We will share results during the discussion period. We note that self-supervised encoders like DINOv2 lack a text encoder, making the gap property inapplicable.
>
> [1] Sun et al., EVA-CLIP, arXiv 2303.15389, 2023.

---

> > ### Author Rebuttal · Reviewer_E4nS · 2026-04-03
> >
> > Most of my concerns have been adequately addressed. I would still appreciate seeing additional experiments exploring different encoder choices, along with deeper theoretical analysis and interpretation. Overall, I am inclined to increase the score to 5.

---

> > > ### Author Response · Authors · 2026-04-07
> > >
> > > We sincerely thank you for the constructive feedback! We are glad that our last responses addressed most of your concerns. Below, we provide the additional encoder experiments and representation analysis you suggested.
> > >
> > > ---
> > > ## Additional Encoder Experiments
> > >
> > > Following your suggestion, we evaluated TextME with a different encoder beyond the default CLIP (ViT-H/14) for image modality used in the main paper. We chose EVA-CLIP (EVA02-E-14-plus) because it represents a distinct encoder family that incorporates masked image modeling during pretraining, allowing us to test whether our centering-based alignment generalizes beyond standard contrastive-only-trained encoders.
> > >
> > > | Model | Flkr. R@1 | Flkr. R@5 | Flkr. R@10 | COCO R@1 | COCO R@5 | COCO R@ 10 |
> > > |:--|---|---|---|---|---|---|
> > > | **Unpaired baselines** | | | | | | |
> > > | Naive | 0.04 | 0.60 | 1.04 | 0.01 | 0.11 | 0.19 |
> > > | COX | 0.20 | 0.20 | 1.00 | 0.02 | 0.14 | 0.26 |
> > > | **Open CLIP-H/14** | | | | | | |
> > > | Pretrained | 77.7 | 94.2 | 96.6 | 48.3 | 73.4 | 82.9 |
> > > | TextME | 51.7 | 77.9 | 86.0 | 28.6 | 54.6 | 66.1 |
> > > | PPR (%) | 66.5 | 82.7 | 89.0 | 59.2 | 74.4 | 79.7 |
> > > | **EVA-02-CLIP-E/14+** | | | | | | |
> > > | Pretrained | 76.1 | 92.6 | 96.1 | 46.2 | 70.5 | 79.3 |
> > > | TextME | 39.1 | 68.5 | 78.1 | 20.6 | 43.3 | 54.3 |
> > > | PPR (%) | 51.4 | 74.0 | 81.3 | 44.6 | 61.4 | 68.5 |
> > >
> > > The results show that TextME successfully transfers to EVA-CLIP, suggesting that it generalizes across diverse encoders. EVA-CLIP shows lower PPR than CLIP, but still achieves 51-81% PPR across benchmarks without any paired supervision, significantly outperforming all unpaired baselines (i.e., Naive and COX).  Below we explain the PPR drop by EVA-CLIP through its geometric properties.
> > >
> > > **Geometric Properties of Pretrained Encoders**
> > >
> > > | Encoder | Modality | (i) Intra-modal indep. | (ii) Gap consistency | (iii) Bounded deviation | (iv) Gap-content ortho. |
> > > |:--|:--|---|---|---|---|
> > > | CLIP ViT-H-14 | Image | 0.28 ($\pm$ 0.11) | 0.97 ($\pm$ 0.00) | 0.00 ($\pm$ 0.00) | 0.00 ($\pm$ 0.11) |
> > > | EVA-02-CLIP-E/14+ | Image | 0.37$^\dagger$ ($\pm$ 0.07) | 0.95$^\dagger$ ($\pm$ 0.01) | -0.01 ($\pm$ 0.09) | 0.00 ($\pm$ 0.07) |
> > >
> > > $\dagger$: weaker satisfaction (>0.1 for (i), <0.96 for (ii)).
> > >
> > >
> > > EVA-CLIP exhibits slightly weaker satisfaction of (i) and (ii) compared to CLIP, which complements the analysis in Section 4.3 by showing that these properties also play a meaningful role in determining alignment quality.
> > >
> > > **Representational Loss Quantification**
> > >
> > > As you suggested in the initial review, we also directly quantified representation-level preservation in Centered Kernel Alignment (CKA) [1] for both encoders to complement the task-level comparison above.
> > >
> > > | Modality | **Gap (orig → proj)** | **CKA** |
> > > |:--|:--:|:--:|
> > > | Image (CLIP) | 0.299 → **0.438** (+46%) | 0.846 |
> > > | Image (EVA-CLIP) | 0.261 → **0.410** (+57%) | 0.831 |
> > >
> > > The CKA scores reveal a consistent pattern with task-level performance. EVA-CLIP shows lower CKA and lower PPR than CLIP, supporting CKA as a useful diagnostic for predicting encoder compatibility with TextME, complementing the geometric analysis above.
> > >
> > > [1] Similarity of neural network representations revisited, ICML 2019.
> > >
> > > ---
> > > ## Theoretical Interpretation
> > >
> > > We interpret these results by building on the theoretical foundations of Zhang et al. (2024a). Their Proposition 1 decomposes paired embeddings as $e_x - e_t = c_\perp + \epsilon$, where $c_\perp$ is a constant gap vector orthogonal to the embedding span and $\epsilon \sim \mathcal{N}(0, \sigma^2 I)$ is alignment noise. Also, their Lemma 1 proves that contrastive gradients lie entirely within the embedding span, meaning the ineffective dimensions where the gap resides receive no gradient update during optimization. This provides the theoretical basis for why TextME’s centering can remove $c_\perp$ while preserving semantic content, with the residual correction reducing to $\epsilon$.
> > >
> > > The four geometric properties measured in Section 4.3 can be understood as empirical diagnostics for how well a given encoder satisfies the assumptions underlying this decomposition. Properties (i) and (ii) assess whether a single constant $c_\perp$ adequately characterizes the modality gap, while properties (iii) and (iv) assess whether offset correction preserves semantic relationships. When these properties are well-satisfied, the residual $\epsilon$ remains small and the theoretical assumptions hold closely, leading to higher downstream performance. We consider this connection between Zhang et al.’s formal results and our empirical geometric analysis to provide a principled account of when and why TextME’s offset-based alignment is effective.
> > >
> > > As we discussed in our initial rebuttal, formally extending these guarantees to modality expansion without paired supervision remains an open theoretical question, and we plan to explore this direction in future work.

---

### Official Review · Reviewer_sii7 · 2026-03-13

**Soundness:** 3
**Presentation:** 3
**Significance:** 3
**Originality:** 3
**Overall Recommendation:** 4
**Confidence:** 4

**Summary:**

This paper addresses the challenge of modality expansion and proposes the TextMe framework, which relies exclusively on textual data. The framework achieves competitive performance using only a modest amount of data and lightweight training.

**Compliance With Llm Reviewing Policy:**

Affirmed.

**Key Questions For Authors:**

Refer to Weakness.

**Limitations:**

The paper lacks an analysis of its limitations, particularly regarding whether it has any limitations compared to LanguageBind.

**Strengths And Weaknesses:**

Strengths
1. The paper evaluates six different modalities, which lends credible support to the generalization capability of the proposed method.
2. The method is clearly presented, requires only lightweight training on text data, and still delivers competitive performance.

Weakness
1. The choice of LanguageBind as the text encoder lacks justification. It is unclear why the authors did not adopt modality-paired text encoders (e.g., using CLIP’s text encoder for image-related tasks).
2. The description of the training dataset appears only at the very end of the Experiments section; it should be moved to the beginning for better flow. Moreover, the dataset consists solely of general text, and the paper does not analyze how incorporating more modality-specific textual data would affect model performance.
3. In the cross-modal (X→X) experiments, TextMe performs significantly worse on the A→I task but markedly better on the 3D→I task, without any explanation or discussion of these contrasting results.
4. The paper lacks t-SNE visualizations showing the embeddings before and after centering, as well as before and after training.
5. Case studies demonstrating the behavior of baseline models should be included.

---

> ### Author Rebuttal · Authors · 2026-03-31
>
> We sincerely thank the reviewer for the thorough feedback. We respond to each point below.
>
> ---
>
> ## W1. Text encoder choice
>
> We chose LanguageBind because it is already fine-tuned from CLIP ViT-L/14 for multimodal expansion across 5 modalities, making it well-suited for our modality expansion setting. Using modality-paired text encoders (e.g., CLAP's text encoder for audio, CXR-CLIP's text encoder for X-ray) would require training a separate projection per text encoder, losing the benefit of a single shared text encoder that enables cross-modal retrieval between arbitrary modality pairs.
>
> To show that the text encoder is not limited to LanguageBind, we replaced it with CLIP ViT-H-14:
>
> | Text Encoder | AudioCaps R@1 | ESC-50 | MN40 | RSNA | DrugBank |
> |:--|---|---|---|---|---|
> | LanguageBind (768d) | 14.5 | 74.3 | 98.8 | 45.0 | 44.1 |
> | CLIP ViT-H-14 (1024d) | 15.9 | 86.7 | 97.9 | 48.3 | 51.7 |
>
> Both produce competitive results across tasks, though CLIP ViT-H-14 shows stronger performance on several benchmarks, which we attribute to its larger model capacity (1024D vs 768D) and broader pretraining. Importantly, TextME works with both encoders without any modification to the framework, confirming its compatibility across text encoder families. We plan to include additional text encoders in the revision to further validate this.
>
> ---
>
> ## W2. Training dataset analysis
>
> We will move the dataset description earlier in the Experiments section for better flow. To deepen the analysis, we measured the distributional distance between each modality-specific corpus and general text (100K samples from Wiki1M [1]) using Maximum Mean Discrepancy (MMD) [2] with RBF kernel in Qwen3-4B embedding space. The table below reports MMD alongside downstream performance when training with modality-specific text vs. Wiki1M, where $\Delta$ denotes the relative improvement of modality-specific text over Wiki1M, computed as $(\text{Modality-specific} - \text{Wiki1M}) / \text{Wiki1M} \times$ 100%. This extends the analysis in Table 5, which compares domain captions against all-NLI; here we use Wiki1M as a more representative general-purpose baseline:
>
> | Modality | MMD | Domain-specific | Wiki1M | $\Delta$ |
> |:--|:--:|:--:|:--:|:--:|
> | Image (Flickr R@1) | 0.245 | 51.66 | 43.16 | 19.7% |
> | 3D (ScanObj Acc) | 0.285 | 42.15 | 34.15 | 23.4% |
> | Audio (AudioCaps R@1) | 0.393 | 15.35 | 5.67 | 170.7% |
> | Molecule (DrugBank MRR) | 0.377 | 26.27 | 9.32 | 181.9% |
>
> Modalities with higher MMD (Audio 0.393, Molecule 0.377) show more than 170% performance gain when switching from Wiki1M to modality-specific text, while lower MMD (Image 0.245, 3D 0.285) shows about 25% gain. This correlation is strong (Pearson $r$ = 0.97, $p$ = 0.03), directly addressing the reviewer's question. Incorporating modality-specific text significantly improves performance for specialized domains, and MMD serves as a practical guideline for practitioners to assess whether modality-specific text is necessary before training.
>
> [1] Gao et al., SimCSE: Simple Contrastive Learning of Sentence Embeddings, EMNLP, 2021.
> [2] Gretton et al., A Kernel Two-Sample Test, JMLR, 2012.
>
> ---
>
> ## W3. Cross-modal A→I vs 3D→I asymmetry
>
> We attribute the difference to encoder geometric properties. Specifically, orthogonality variance (property (iv) in Section 4.3) measures how consistently the modality gap direction remains independent of semantic content across individual instances. Uni3D has low orthogonality variance ($\pm$ 0.04), leading to precise offset correction, which is why 3D→I outperforms Ex-MCR (10.27 vs 5.67 in R@1). In contrast, CLAP has higher orthogonality variance ($\pm$ 0.15), introducing greater distortion in the Audio projection, resulting in lower A→I performance (TextME 1.06 vs Ex-MCR 1.57 in R@1). Cross-modal retrieval is particularly sensitive to this because both the query and target modality must be independently projected into LLM space (e.g., Audio→LLM and Image→LLM), accumulating errors from both projections. We will add this discussion to the revision.
>
> ---
>
> ## W4 & W5. t-SNE visualizations and baseline case studies
>
> We have generated t-SNE visualizations for all 5 modalities showing three stages of the TextME framework (before centering, after centering, after projection), along with baseline case studies (Naive, COX, Ex-MCR) on the same cross-modal retrieval tasks. Both can be viewed at https://anonymous.4open.science/r/textme-rebuttal-figures-C568/. The visualizations show that for modalities with strong geometric properties (Audio, Image), text and modal embeddings progressively overlap after centering and become well-interleaved after projection, while for modalities with weaker geometry (Molecule), the alignment effect is more subtle. We will add the baseline case studies to Figure 4 in the revision.

---

> > ### Author Rebuttal · Reviewer_sii7 · 2026-04-03
> >
> > The authors' response has addressed my concerns. I will maintain my original score.

---

> > > ### Author Response · Authors · 2026-04-07
> > >
> > > We thank you for the careful review and are glad that our responses addressed your concerns. We will incorporate all your suggestions, including the t-SNE visualizations and baseline case studies, in the revised manuscript.

---

### Official Review · Reviewer_isru · 2026-03-17

**Soundness:** 3
**Presentation:** 3
**Significance:** 3
**Originality:** 2
**Overall Recommendation:** 4
**Confidence:** 2

**Summary:**

This paper proposes a text-only modality expansion framework called TextME, which leverages the geometric "modality gap" property of pretrained
  encoders to project multiple different modalities into a unified large language model embedding space using only unpaired text descriptions. By
  completely removing the need for expensive paired multimodal data, the method greatly reduces training cost while successfully enabling zero-shot
  cross-modal retrieval and classification.

**Compliance With Llm Reviewing Policy:**

Affirmed.

**Ethical Review Concerns:**

None.

**Final Justification:**

My concerns have been fully resolved.

**Key Questions For Authors:**

- Novelty of the core mechanism: The paper uses centering to remove modality gaps and build an interchangeable space, but similar geometric
    alignment has already been explored in prior work such as Zhang et al., 2024a under paired-data settings. Is TextME’s main novelty therefore its
    extension to the no-paired-data setting, or does it offer a deeper algorithmic or theoretical advance beyond using LLM embeddings as a shared
    anchor?
  - Dependence on domain-specific text: The experiments show that domain-specific captions greatly outperform general corpora like all-NLI, with 3D
    performance dropping from 70.86% to 12.10%. In specialized areas such as medical imaging or molecular data, is collecting 100,000 high-quality in-
    domain text descriptions still a major practical bottleneck?
  - Limits on specialized modalities: The molecular modality shows weak gap consistency at 0.78, and offset correction even reduces performance by
    4.6%, with retrieval retention only around 42%. Does this mean TextME is fundamentally limited when pretrained encoders lack the desired geometry,
    and is there a fallback strategy for such cases?
  - Anchor space bottleneck: Although Qwen3-Embedding is chosen as the default anchor, ablations show that multimodal encoders like CLIP or
    LanguageBind can outperform it on classification. Could relying on a single LLM anchor limit classification performance, and might combining
    multiple anchor spaces be a better direction?

**Limitations:**

Yes.

**Strengths And Weaknesses:**

Strengths

  - Low compute and training cost: The framework reuses frozen pretrained contrastive encoders and trains only a lightweight projection network of
    about 10M parameters per modality.
  - Emergent cross-modal capability: By using a unified LLM anchor space, the framework enables direct retrieval between modality pairs never seen
    during training, such as audio-to-3D and molecule-to-image.
  - Deep mechanistic analysis: The paper goes beyond benchmark results by quantifying geometric properties such as gap-content orthogonality and gap
    consistency to explain why text-only expansion works.

  Weaknesses & Limitations

  - Unstable performance across tasks and modalities: Performance retention is much lower for retrieval tasks requiring fine-grained instance-level
    similarity, with molecular retrieval retaining only about 42%.
  - Strong dependence on encoder geometry: The method relies heavily on pretrained encoders having strong gap consistency and low-variance gap-content
    orthogonality, and it can fail when these properties are weak.
  - Sensitivity to training text distribution: The experiments show that using general natural language inference corpora like all-NLI causes a
    substantial performance drop, indicating dependence on domain-matched text descriptions.
  - Potential bias in offset estimation: The paper notes that the semantic distribution of the dataset used to compute modality offsets may affect the
    resulting offset vectors and downstream alignment quality.

---

> ### Author Rebuttal · Authors · 2026-03-31
>
> We thank the reviewer for the detailed feedback. We address each point below with new experiments.
>
> ---
>
> ## Q1. Novelty of the core mechanism
>
> TextME is, to the best of our knowledge, the first framework that expands independently pretrained encoders to new modalities using only text descriptions for projection training, without any paired multimodal supervision. This setting is distinct from Zhang et al. (2024a), which operates centering within a single bimodal encoder’s shared embedding space (e.g., CLIP), and is not designed for bridging multiple independently pretrained encoders with disjoint embedding spaces.
>
> TextME extends beyond this setting by (1) bridging heterogeneous embedding spaces from independently pretrained encoders (e.g., CLAP, Uni3D, MoleculeSTM) that were never jointly trained, (2) exploiting text as a shared modality across all multimodal representation spaces and introducing LLM embedding as a unified anchor to connect their disjoint representations, and (3) verifying the gap property across 6 diverse modality encoder families, revealing that it does not hold uniformly across encoders (e.g., molecule gap consistency = 0.78) and establishing a quantitative diagnostic (Section 4.3, Figure 5) for predicting when text-only projection learning will succeed.
>
> ---
>
> ## Q2. Dependence on domain-specific text & W3. Sensitivity to training text distribution
>
> We clarify that "domain-specific text" refers to unpaired text descriptions, not (modality, text) pairs. Such domain-relevant descriptions already exist at scale in public repositories (e.g., PubChem 119M compounds, Objaverse 800K+ 3D captions, MIMIC-CXR 227K radiology reports). Moreover, LLM-synthesized descriptions can replace domain-specific paired dataset collection, achieving 86% of original performance while outperforming on classification.
>
> | Training Data | AudioCaps R@1 | ESC-50 Acc |
> |:--|:--:|:--:|
> | Domain captions | **14.5** | 74.3 |
> | **Synthesized (LLM-gen.)** | 12.5 | **79.2** |
> | Wiki1M (general) | 7.3 | 62.0 |
>
> Therefore, we believe that the practical burden of obtaining domain-relevant text is substantially lower than paired data collection required by prior methods.
>
> ---
>
> ## Q3. Limits on specialized modalities & W2. Strong dependence on encoder geometry & W4. Potential bias in offset estimation
>
> We agree that TextME's effectiveness depends on encoder geometry. However, our core contribution lies in leveraging these properties to enable text-only projection learning with a single global offset. This is grounded in Lemma 1 of Zhang et al. (2024a), which proves that contrastive optimization preserves the modality gap orthogonal to the embedding span, ensuring that centering removes only modality-specific bias without distorting semantic content. The encoders we adopt empirically satisfy this gap structure (Table 3), validating the use of a single global offset.
>
> For the fallback strategy, we also tested whether the single global offset can be improved by capturing local structure or non-linearity in the modality gap, focusing on molecule and audio (highest orthogonality variance, Table 2). For local structure, multi-cluster offset ($K=7$) partitions the 5K samples via k-means and applies a separate offset per cluster. For non-linearity, a learned 2-layer MLP trained on 5K paired samples replaces the fixed offset with a nonlinear mapping.
>
> | Method | Mol MRR | Audio R@1 |
> |:--|:--:|:--:|
> | No offset | 36.4 | 8.7 |
> | Global offset (K=1, unpaired) | 34.8 | 15.4 |
> | Multi-cluster offset (K=7, unpaired) | 34.8 | 12.5 |
> | Learned MLP offset (paired) | 33.1 | 15.3 |
>
> As a result, neither outperforms the global offset. For context, pretrained CLAP achieves 22.47 in R@1 using over 2M paired samples, while TextME preserves 68.3% of this performance with only 5K unpaired samples for offset. Even if a local or non-linear structure exists in the modality gap, capturing it from such limited unpaired data remains an open challenge that we plan to investigate further.
>
> ---
>
> ## Q4. Anchor space bottleneck & W1. Unstable performance across tasks and modalities
>
> Many thanks for the great suggestion. Following your suggestion, we explored a multi-anchor ensemble via score-level late fusion, aggregating scores from 4 anchors at inference time.
>
> | **Anchor** | **AudioCaps R@1** | **DrugBank Acc@10** | **MN40 Acc** | **ESC-50 Acc** | **RSNA Acc** |
> |:--|---|---|---|---|---|
> | CLIP (1024d) | 15.9 | 36.4 | 78.0 | **86.7** | 48.3 |
> | LanguageBind (768d) | 14.5 | 29.7 | **81.1** | 74.7 | 45.0 |
> | NV-Embed-v2 (4096d) | 16.2 | 26.3 | 78.3 | 79.4 | **48.6** |
> | Qwen3-Embed (2560d) | 15.4 | 34.8 | 79.0 | 77.2 | 46.6 |
> | **Ensemble** | **19.2** | **41.5** | 79.7 | **86.7** | 46.9 |
>
> Ensemble improves retrieval by combining complementary signals, directly addressing the retrieval-classification gap. This observation, i.e., a simple ensemble of multiple models could effectively mitigate such a bottleneck, is consistent with our claim (lines 297-302).

---

> > ### Author Rebuttal · Reviewer_isru · 2026-04-04
> >
> > Thank you for your response. My concerns have been fully resolved.

---

> > > ### Author Response · Authors · 2026-04-07
> > >
> > > We sincerely thank you for the constructive review and are glad that our responses fully addressed your concerns. Your feedback has been valuable in strengthening the paper, and we will incorporate all the suggested improvements in the revised manuscript.

---

### Decision · Program_Chairs · 2026-04-30

**Decision:**

Accept (regular)

**Comment:**

This paper was reviewed by four experts and received two Weak Accept, one Accept, and one Weak Reject rating. Based on the reviews, rebuttal, and discussion with the SAC, I recommend the paper for acceptance to ICML 2026. Reviewers viewed the paper positively overall, highlighting the importance of the problem, the simplicity and efficiency of the proposed approach, and the breadth of the empirical evaluation across six modalities and multiple task types. The paper's central idea is interesting and well-motivated: exploiting the modality-gap geometry of pretrained contrastive encoders to train lightweight modality-specific projectors using only text descriptions, enabling alignment of diverse modalities into a shared LLM embedding space without new paired multimodal supervision at the projector-training stage. The rebuttal further strengthened the paper with useful analysis on encoder choice, representation preservation, anchor-space behavior, and the effect of more expressive offset corrections, and resolved most of the reviewers' concerns.

In the final version, the authors advised to include the following points based on the rebuttal discussion:  (1) revise terminology around "text-only" (as the authors agreed, "text-only projection learning" would be more accurate); (2) clarify positioning relative to prior work on modality-gap centering to reflect "text-only projection learning"; (3) discuss practical limitations, including performance on fine-grained retrieval and dependence on favorable encoder geometry.

Overall, the paper makes a useful and timely contribution to multimodal representation learning that is likely to be of broad interest. We congratulate the authors on the acceptance of their paper.